# Perceptual fusion of musical notes by native Amazonians suggests universal representations of musical intervals

Malinda J. McPherson [1,2,3✉], Sophia E. Dolan[4], Alex Durango[1,3], Tomas Ossandon [5,6], Joaquín Valdés [5], Eduardo A. Undurraga [7,8], Nori Jacoby[9], Ricardo A. Godoy[10] & Josh H. McDermott [1,2,3,11✉]

Music perception is plausibly constrained by universal perceptual mechanisms adapted to natural sounds. Such constraints could arise from our dependence on harmonic frequency spectra for segregating concurrent sounds, but evidence has been circumstantial. We measured the extent to which concurrent musical notes are misperceived as a single sound, testing Westerners as well as native Amazonians with limited exposure to Western music. Both groups were more likely to mistake note combinations related by simple integer ratios as single sounds ('fusion'). Thus, even with little exposure to Western harmony, acoustic constraints on sound segregation appear to induce perceptual structure on note combinations. However, fusion did not predict aesthetic judgments of intervals in Westerners, or in Amazonians, who were indifferent to consonance/dissonance. The results suggest universal perceptual mechanisms that could help explain cross-cultural regularities in musical systems, but indicate that these mechanisms interact with culture-specific influences to produce musical phenomena such as consonance.

[1] Department of Brain and Cognitive Sciences, MIT, Cambridge, MA 02139, USA. [2] Program in Speech and Hearing Biosciences and Technology, Harvard University, Boston, MA 02115, USA. [3] McGovern Institute for Brain Research, MIT, Cambridge, MA 02139, USA. [4] Wellesley College, Wellesley, MA 02481, USA. [5] Department of Psychiatry, School of Medicine, Pontificia Universidad Católica de Chile, Santiago, Chile. [6] Institute for Biological and Medical Engineering, Schools of Engineering, Medicine and Biological Sciences, Pontificia Universidad Católica de Chile, Chile. [7] Escuela de Gobierno, Pontificia Universidad Católica de Chile, Santiago, Chile. [8] Millennium Nucleus for the Study of the Life Course and Vulnerability (MLIV), Santiago, Chile. [9] Max Planck Institute for Empirical Aesthetics, Frankfurt, Germany. [10] Heller School for Social Policy and Management, Brandeis University, Waltham, MA 02453, USA. [11] Center for Brains, Minds and Machines, MIT, Cambridge, MA 02139, USA. ✉email: mjmcp@mit.edu; jhm@mit.edu

Music is present in every culture, and some features of musical systems and behavior are widespread[1–3]. These cross-cultural regularities could reflect biological constraints, as might arise from universal perceptual mechanisms adapted to natural sounds[4,5]. But due to the dearth of cross-cultural perception research, little is definitively known about universal mechanisms of hearing that might serve as such constraints on musical systems.

One prominent regularity in natural sounds is harmonicity—the presence of frequencies (harmonics) that are integer multiples of a common fundamental frequency (f0). Harmonic frequencies provide a key acoustic grouping cue allowing listeners to segregate concurrent sound sources[6–8] such as speech[9], and to attend to particular sound sources in auditory scenes[10,11]. The absolute f0 associated with a set of harmonics is also critical for recognizing voices[12,13]. Moreover, neuronal selectivity for harmonic frequencies is evident in at least one species of non-human primate[14]. The importance of harmonicity in perceptual contexts that are presumably shared across all human societies suggests that mechanisms related to harmonicity could provide a strong constraint on musical behavior.

Scholars have long noted the potential relationships between the structure and perception of Western music and the harmonic series, particularly for musical harmony[15–29]. Pairs of notes whose f0s are related by simple integer ratios, and whose combined spectra approximate the harmonic frequencies of a single note, are generally regarded as consonant by Westerners (Fig. 1a). For Western listeners, converging evidence supports some relationship between harmonicity and consonance—here operationalized as the pleasantness of note combinations[24–27,29]. However, the extent to which consonance preferences in Westerners can be fully predicted by similarity to the harmonic series is unknown, in part because we lack widely accepted models for how harmonicity is represented in the auditory system[23,25,30]. In particular, the perception of harmonicity is strongly influenced by the position of harmonics within the harmonic series[30], and is not well captured by naive measures based on harmonic templates or autocorrelation. Moreover, in contrast to the presumptive worldwide perceptual importance of harmonicity, consonance preferences vary across cultures, and in some cases appear to be absent[31]. And even within Western cultures, definitions of consonance have fluctuated over history[15], and preferences for consonance vary with musical experience[24,29]. Whether consonance preferences (when present) reflect in some way universal perceptual constraints has thus remained unclear.

One other potential consequence of harmonicity is the phenomenon of fusion—the tendency of note combinations related by simple integer ratios (whose spectra approximate the harmonic series, Fig. 1a) to resemble a single sound[18]. Although fusion is a widely acknowledged phenomenon that has been cited as a feature of consonance for centuries[15,18,32–34], measurements of fusion for musical note combinations have been rare. We are aware of only one modern set of psychophysical measurements of fusion[35], and that study used a complex method in which additional notes were added to the pitch intervals of interest. Some reason to think that consonance might not be rigidly tied to fusion comes from the small deviations from integer ratios imposed by modern tuning systems (e.g., equal temperament), that might be expected to impair fusion[35] despite apparently having minimal effect on consonance judgments. Others have also questioned the relationship between consonance and fusion based on informal musicological observation[36]. However, such discussions have been limited by the lack of data relating the two phenomena.

Irrespective of its relationship to consonance, harmonicity-based sound segregation might provide an important constraint

on musical systems, particularly if its perceptual effects on musical note combinations were present cross-culturally. A priori, it seemed plausible that this might be the case, but not inevitable. The measurements of fusion that exist have been limited to Western listeners, who have extensive exposure to harmony featuring simple-integer-ratio intervals, and for whom fusion could thus reflect learned schemas[4,37], potentially incorporating the idiosyncrasies of modern tuning systems. Moreover, sound segregation abilities are often thought to change with musical training[38] (though see refs. [39,40]), which might suggest that the phenomenon of fusion could vary across cultures differing in their musical experience. To address these outstanding issues we sought to verify the phenomenon of fusion in Westerners, test its robustness to tuning systems, test its relation to consonance in Westerners, and explore the extent to which it is present cross-culturally.

We conducted cross-cultural experiments to assess fusion of musical note combinations along with aesthetic responses to the same stimuli. The Tsimane', a small-scale native Amazonian society in Bolivia[41] (Fig. 1b), are an interesting group in which to assess fusion because they have limited exposure to Western music, and because group performances have traditionally been absent from their culture[31,42]. As a result, the Tsimane' appear to have little experience with concurrent pitches in music (see "Background information on Tsimane' music", in "Methods"). Moreover, recent experiments indicate that they lack the preference for consonant over dissonant chords that is typically present in Western listeners[31]. In that earlier study we found that the Tsimane' could detect mistuned harmonics[31], suggesting some sensitivity to harmonicity, but their overall sensitivity was worse than the comparison group of Westerners, and the experiment did not involve actual musical intervals (pairs of notes). It was thus unclear whether their perceptual representations of musical intervals (as measured by fusion) would qualitatively differ along with their aesthetic evaluations, or whether their representations of note combinations would resemble those of Westerners despite these differences in aesthetic evaluations.

## Results

**Overview of experiments.** On each trial of the main fusion experiments, listeners heard a stimulus (two concurrent notes separated by a particular musical interval; Figs. 1c and 2a) and judged whether it contained one or two sounds. It seemed plausible that in participants without much experience with musical harmony, the fusion of a musical interval might simply decrease with increasing pitch difference between the notes. Intervals were thus selected to range from small (major second) to large (minor ninth) pitch differences, with dissonant and consonant intervals (as judged by Westerners) intermingled when ranked by size (Fig. 1d). The major second, tritone, major seventh, and minor ninth, all considered dissonant by Western listeners, alternated with the third, fourth, fifth, and octave, all defined by simple integer ratios and considered consonant. The predictions of fusion based on similarity to the harmonic series (via simple integer ratios, as are related to Western consonance) were thus differentiated from those of the size of the pitch difference between notes.

One set of experiments was conducted with synthetic tones. For these experiments, two different tuning systems were used (Supplementary Table 1), in separate blocks. We aimed to test whether the small deviations from harmonicity present in modern instrument tuning might affect perceptual equivalences[35], and whether any such effect might interact with musical experience. The intervals tested differed by 0–13.7 cents between the two tuning systems, depending on the interval

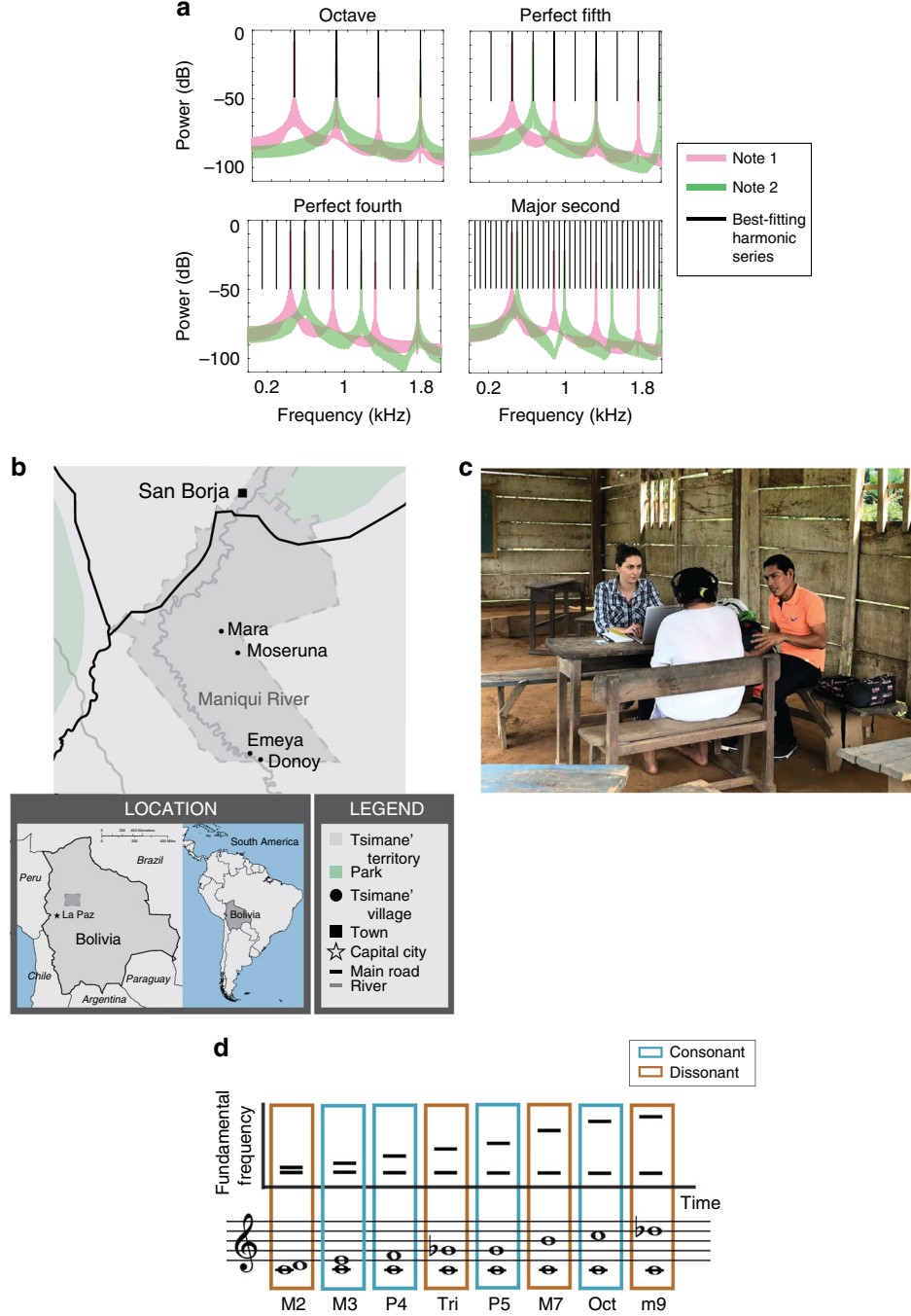

**Fig. 1 Stimuli and experimental setup. a** Frequency spectra of example musical interval stimuli, each generated from the superposition of two harmonic notes separated by a particular pitch interval. Canonically consonant intervals such as the octave, perfect fifth, and perfect fourth produce aggregate spectra whose frequencies approximate harmonics of a common fundamental frequency (f0). The octave has the greatest overlap with the best matching harmonic series, followed by the fifth, and then the fourth. Canonically dissonant intervals such as the major second produce an aggregate set of frequencies that are consistent only with a much lower f0, and that produce little overlap with its harmonics. **b** Map of the region of Bolivia where testing occurred. Tsimane' participants were residents of the villages of Mara, Moseruna, Emeya, and Donoy. **c** Example of experimental conditions; photo depicts an experiment conducted in a classroom. Sounds were presented over closed headphones using a laptop, and were audible only to the participant (such that the experimenter was blind to the stimulus being presented). A translator (shown here with the orange shirt) assisted in explaining the experiments and interpreting responses. **d** Schematic and musical depiction of consonant and dissonant musical intervals used in experiments, showing alternation of consonant and dissonant intervals as interval magnitude (difference between the note f0s) increases.

(Supplementary Table 1). For most analyses we combined results across the two tuning systems, as we never observed significant differences between responses to stimuli in the two tuning systems in any of our experiments.

To assess whether the results might extend to more naturalistic musical stimuli, we conducted an analogous experiment using sung notes. In order to assess perceptual abilities in the absence of training, and to make the fusion experiments comparable to the

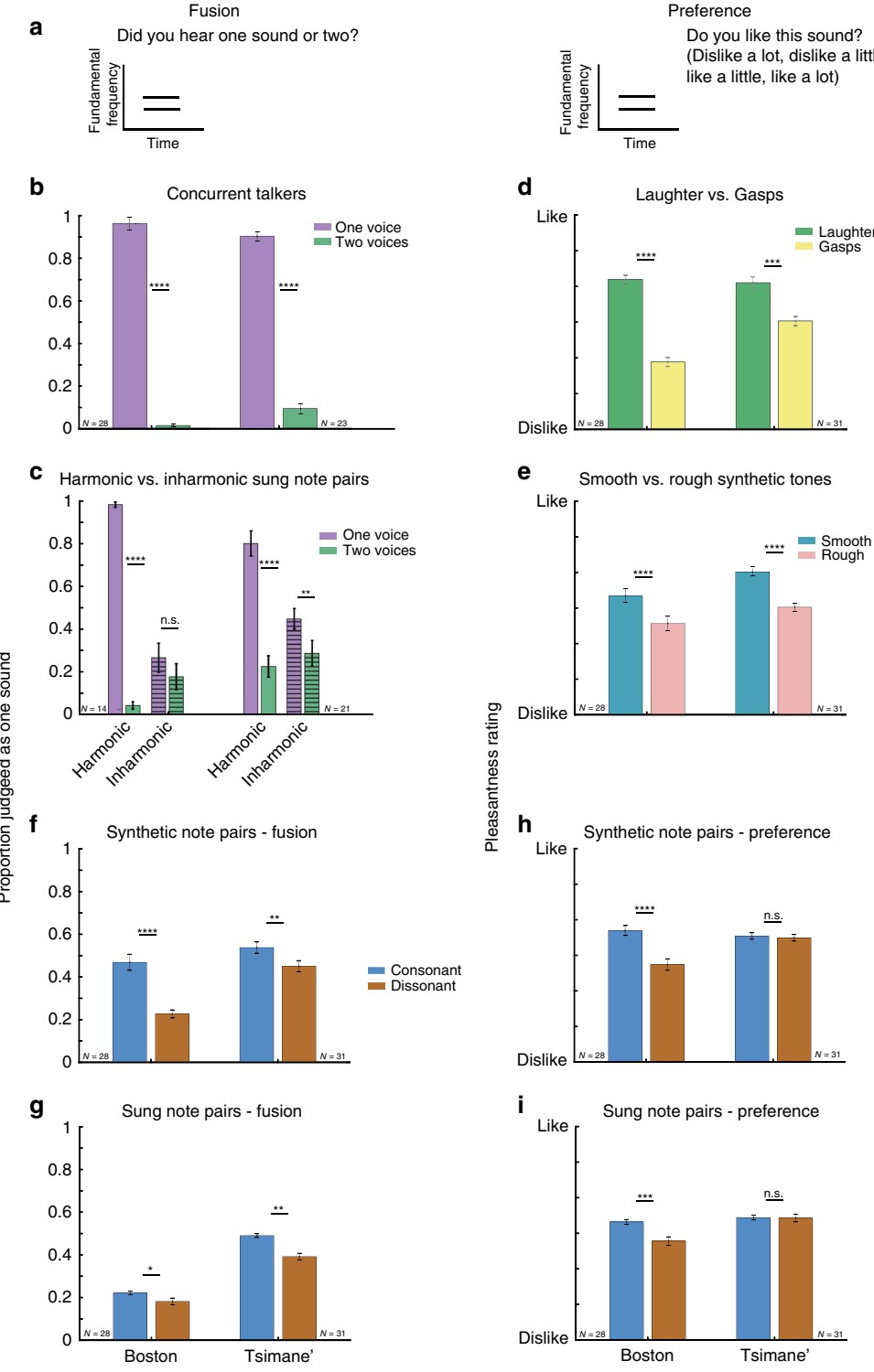

pleasantness rating experiments used to assess consonance, no feedback was given in any of the experiments.

To help confirm task comprehension and compare the mechanisms of sound segregation across cultures, we conducted two control experiments for the fusion task. In the first, listeners heard segments of Tsimane' speech—either a single person talking, or two people talking concurrently. In a second experiment with different participants, listeners heard one or two concurrent sung vowels that were resynthesized to be either harmonic or inharmonic[9]. We expected Westerners to be better at determining the number of voices for harmonic compared to

inharmonic signals, and sought to test whether Tsimane' listeners would exhibit a similar advantage.

Listeners also performed a pleasantness rating task on the same musical interval stimuli used in the fusion experiments, as well as two other control experiments to validate comprehension of the pleasantness rating task. These control experiments featured rough and smooth synthetic tones, and recorded vocalizations with positive and negative valence (laughter and gasps).

We ran identical sets of experiments on two participant groups: US non-musicians residing in the Boston metropolitan area and Tsimane' residing in villages in the Amazon rain forest. Tsimane'

**Fig. 2 Summary of results of in-person experiments. a** Schematic of trial structure for fusion (left) and preference (right) experiments. Participants heard a stimulus, and judged whether it contained one or two sounds, or rated its pleasantness. Line segments denote individual notes, as were presented in the main experiments with musical intervals. **b** Results of first fusion control experiment, in which participants heard one or two concurrent talkers. Not all Tsimane' participants completed the experiment (hence the smaller $N$ compared to other panels). Here and in (**c**), (**f**), and (**g**), graph plots proportion of trials on which participants reported hearing one sound, plotted separately for Boston non-musicians and Tsimane'. Here and in (**c**)–(**f**) and (**h**), plots show the mean ± SEM. Results for individual participants are shown in Supplementary Fig. 1. **c** Results of second fusion control experiment, in which a separate set of participants heard one or two concurrent sung vowels, resynthesized to be either harmonic or inharmonic. F0 difference between vowels was chosen to avoid fusion in Western listeners when the notes were harmonic. Participants for this experiment were different from those for other experiments (hence different sample size). **d**, **e** Results of preference control experiments, in which participants rated the pleasantness of recorded laughs and gasps, and of smooth and rough synthetic tones, respectively. In the latter case, tones consisted of pairs of frequencies presented either dichotically, to avoid beating, or diotically, to produce beating (roughness). **f** Results of fusion experiment with musical intervals composed of synthetic notes. Fusion judgments were pooled across canonically consonant and dissonant musical intervals (and across tuning systems, which gave indistinguishable results). **g** Results of fusion experiment with sung notes, pooled across consonant and dissonant intervals. Here and in (**i**), plots show the mean ± within-participant SEM. **h** Results of preference experiments with musical intervals composed of synthetic notes (averaged within consonant and dissonant subsets, and tuning systems). **i** Results of preference experiments with musical intervals composed of sung notes (averaged within consonant and dissonant subsets). Across all results graphs, asterisks denote statistical significance of pairwise comparisons: *$p < 0.05$, **$p < 0.01$, ***$p < 0.001$, ****$p < 0.0001$, and n.s. not significant. For subplots (**b**), (**c**), (**f**), and (**g**), two-sided Wilcoxon signed-rank tests were used to test for differences between conditions. For subplots (**d**), (**e**), (**h**), and (**i**), two-tailed paired $t$-tests were used to test for differences between conditions.

have traditionally lived in small villages without electricity or running water, with limited contact with the rest of Bolivia. Rain makes local roads impassable to vehicles for much of the year, contributing to the region's isolation. The Tsimane' territory is undergoing rapid changes due to efforts by the Bolivian government to expand public health services, schools, and roads to indigenous groups[43], but our testing was restricted to villages that lack electricity and cell phone reception and that remain relatively isolated (accessible from town only by a 2-day walk or 3-hour drive during the dry season, or by a 2-day trip in a motorized canoe, depending on the village).

**Control experiments verify task comprehension**. Both participant groups were well above chance on the first fusion control experiment, in which they reliably categorized sounds as containing one or two talkers (Fig. 2b; individual participant data for all results graphs in Fig. 2 are shown in Supplementary Fig. 1). Non-parametric tests were used to evaluate all fusion experiments as the data were non-normal. The difference between one- and two-talker conditions was highly significant for both groups ($Z = 4.87$, $p < 0.0001$ for Boston, $Z = 4.22$, $p < 0.0001$ for Tsimane', Wilcoxon signed-rank test); when expressed as sensitivity, Boston participants achieved a mean $d$-prime of 4.27, and Tsimane' participants achieved a mean $d$-prime of 3.23. This result suggests that participants in both groups understood the task of discriminating one vs. two concurrent sounds.

In the second control experiment with concurrent sung vowels, US and Tsimane' participants both performed substantially better when the vowels were harmonic than when they were inharmonic (Fig. 2c). The proportion of single voices correctly identified as one voice was higher for harmonic than inharmonic trials, $Z = 3.31$, $p < 0.001$ for Boston, $Z = 3.43$, $p < 0.001$ for Tsimane', as was discrimination between one and two voices (assessed with $d$-prime), $Z = 3.31$, $p < 0.001$ for Boston, $Z = 3.84$, $p < 0.001$ for Tsimane'. This result suggests that both Tsimane' and US listeners make use of harmonic frequency structure when segregating concurrent sounds. These control experiments indicate that basic sound segregation competencies are present across cultures and are at least to some extent reliant on cues related to harmonicity.

The results of the preference control experiments also indicated task comprehension. Preferences were present in both participant groups for positively-valenced over negatively-valenced vocalizations (Fig. 2d; recordings of laughs and gasps), and for smooth over rough synthetic tones (Fig. 2e; pairs of frequencies presented

to either the same or different ears, differing in the sensation of beating). In each experiment the stimulus type produced highly significant effects for each participant group (roughness: $t(27) = 5.21$, $p < 0.0001$, $d = 0.75$ for Boston, $t(30) = 5.59$, $p < 0.0001$, $d = 1.47$ for Tsimane'; vocalizations: $t(27) = 10.76$, $p < 0.0001$, $d = 2.56$ for Boston $t(30) = 4.14$, $p < 0.001$, $d = 1.07$ for Tsimane', two-tailed paired $t$-test and Cohen's $d$). These control experiments verify that any cross-cultural variation in consonance preferences is unlikely to be due to an inability to perform the task or a misunderstanding of the instructions.

**Cross-cultural fusion of consonant intervals**. With musical intervals, Westerners exhibited greater fusion for consonant than dissonant intervals, as expected, but so did the Tsimane'. This was true both for synthetic (Fig. 2f) and sung (Fig. 2g) notes. In both experiments and in both groups, consonant intervals were significantly more fused than dissonant intervals: synthetic notes, Boston ($Z = 4.23$, $p < 0.0001$), Tsimane' ($Z = 3.27$, $p = 0.001$), and sung notes, Boston ($Z = 2.19$, $p = 0.028$), Tsimane' ($Z = 2.83$, $p = 0.005$). Boston participants showed a stronger effect of interval type (consonant vs. dissonant) for synthetic notes (Cohen's $d = 1.41$ for Boston participants vs. $d = 0.69$ for Tsimane'), but Tsimane' showed a stronger effect for sung notes (Cohen's $d = 0.47$ for Tsimane' participants vs. $d = 0.20$ for Boston). Pooling across experiments with synthetic and sung notes, there was no interaction between participant group and interval type ($F(1,57) = 3.42$, $p = 0.07$, $\eta_p^2 = 0.057$, significance evaluated via bootstrap due to non-normality). However, the overall degree of fusion of musical intervals varied across groups —Tsimane' were more likely to mistake two notes for one (main effect of group, pooled across both experiments, $F(1,57) = 25.20$, $p < 0.001$, $\eta_p^2 = 0.31$).

The increased fusion for consonant intervals was independent of the tuning system: we found no interaction between the effects of tuning (just intonation vs. equal temperament) and interval type (consonant vs. dissonant) on fusion ($F(1,57) = 0.04$, $p = 0.83$, $\eta_p^2 = 0.001$, experiment with synthetic notes where both tuning systems were tested; Supplementary Fig. 2). There was also no main effect of tuning system ($F(1,57) = 1.46$, $p = 0.23$, $\eta_p^2 = 0.3$). The similarity of the results across tuning systems indicates that the mechanism underlying fusion is sufficiently coarse that the small differences in frequency ratios imposed by tuning systems do not have much effect, even in a population with presumptively little exposure to equal temperament tuning.

**Fusion dissociates from consonance preferences.** In contrast to the cross-cultural similarity of fusion judgments, preferences for consonant over dissonant intervals varied across groups (Fig. 2h, i), as in previous work[31]. In both experiments (synthetic and sung notes), the preference was robust in US listeners ($t(27) = 6.57$, $p < 0.0001$, $d = 1.21$ for synthetic notes, $t(27) = 4.02$, $p < 0.001$, $d = 0.65$ for sung notes), but undetectable in Tsimane' ($t(30) = 0.58$, $p = 0.57$, $d = 0.10$ for synthetic notes, $t(30) = 0.13$, $p = 0.90$, $d = 0.02$ for sung notes), producing an interaction between the effect of interval type (consonant vs. dissonant) and participant group on pleasantness ratings ($F(1,57) = 26.48$, $p < 0.001$, $\eta_p^2 = 0.32$, mixed model ANOVA with within-subject factors of experiment and consonance vs. dissonance, averaged across synthetic and sung notes). This variation was independent of the tuning system: we found no interaction between the effect of tuning and consonance/dissonance on pleasantness ratings in either group ($F(1,30) = 0.05$, $p = 0.82$, $\eta_p^2 = 0.002$ for Tsimane' participants, $F(1,27) = 3.22$, $p = 0.84$, $\eta_p^2 = 0.11$ for Boston participants, when analyzing the experiments with synthetic notes, for which two tuning systems were employed).

We note that the roughness preference from the second preference control experiment (Fig. 2e) was no larger in Boston participants than the consonance preference: there was no significant difference between the difference in the mean ratings being contrasted in these experiments: rough vs. smooth and consonant vs. dissonant synthetic tones ($t(27) = 0.27$, $p = 0.33$, $d = 0.22$). The roughness preference evident in the Tsimane' thus provides evidence that the apparent lack of consonance preferences in the Tsimane' is not due to a lack of experimental sensitivity.

**Consistency in fused intervals across cultures.** The detailed pattern of fusion across individual intervals also exhibited similarities across groups, particularly for synthetic notes (Fig. 3a, b). In both groups, fusion was greatest for the octave, and the octave, fifth, and fourth all exhibited greater fusion than the dissonant intervals closest in size (Wilcoxon signed-rank tests; US participants: $Z > 3.91$, $p < 0.001$ for all comparisons between the octave, fifth and fourth and the mean of the two adjacent dissonant intervals; Tsimane' participants: $Z > 3.33$, $p < 0.001$ for octave and fourth, $Z = 2.55$, $p < 0.05$ for the fifth; experiments with synthetic notes).

US participants exhibited less fusion overall for sung notes compared to synthetic notes, plausibly due to the additional segregation cues available in actual vocal stimuli (e.g., the distinct pitch modulation within each note, or the different vocal tract filtering of the two notes). Listeners might become additionally sensitive to these cues with experience with vocal harmony[37] (which appears to be uncommon in Tsimane' music). However, cross-cultural similarities in the perception of sung intervals were nonetheless evident, with the octave producing stronger fusion than adjacent intervals in both groups (Fig. 3c, d; $Z = 2.22$, $p = 0.03$ for Boston, $Z = 4.31$, $p < 0.0001$ for Tsimane').

Notably, fusion in the Tsimane' was similar for sung and synthetic stimuli (no significant interaction between the experiment and the effect of musical interval, $F(8,30) = 0.96$, $p = 0.47$, $\eta_p^2 = 0.03$, Fig. 3b, d). In particular, the octave produced the strongest fusion in both experiments. This result is consistent with the idea that fusion could have real-world musical relevance, making the (harmonic) octave perceptually distinctive compared to other intervals.

**Fusion does not predict consonance in Westerners.** Although the consonant intervals in our experiment were overall more fused than the dissonant intervals, the pattern of fusion did not mirror the pattern of consonance preferences at the level of individual intervals (Fig. 3e–h). Even in Westerners, who exhibit robust preferences for consonant intervals, the octave was most fused but not most pleasant. By contrast, the third was rated as highly as other consonant intervals but was less fused (Fig. 3e vs. 3a). And for sung notes, Westerners showed preferences for all canonically consonant intervals even though fusion was much less consistently present (producing an interaction between the type of judgment and the interval; $F(8,216) = 10.71$, $p < 0.001$, $\eta_p^2 = 0.28$ for synthetic notes; $F(8,216) = 12.23$, $p < 0.001$, $\eta_p^2 = 0.31$ for sung notes, significance calculated via bootstrap). Fusion also did not predict preferences in Tsimane' participants, who did not prefer consonant to dissonant intervals, instead showing a slight preference for larger intervals, consistent with previous findings[31].

**Fusion and consonance in a large online cohort of Westerners.** To further assess the relationship between consonance and fusion in Westerners, we conducted an online experiment with Western non-musicians in order to run enough participants ($N = 100$) to obtain highly reliable results at the level of individual musical intervals (Fig. 4a, b). To more exhaustively measure interval fusion, we included a larger set of intervals than in the in-person experiments, ranging from 0 to 14 semitones. As in the in-person experiments, each participant judged each stimulus as one or two sounds (fusion), and rated its pleasantness (here described synonymously as consonance), in separate blocks, the order of which was randomized across participants.

This online experiment replicated and extended the results of the in-person experiment, showing peaks in fusion at the octave, fifth, and fourth, and a pattern of rated pleasantness (consonance) consistent with many prior experiments[24,29]. When directly compared, mean consonance and mean fusion for Westerners were correlated across intervals ($r_s = 0.85$, $p < 0.001$, Spearman rank correlation), as expected given that both are believed to be related to the harmonic series. However, consonance and fusion also exhibited consistent dissociations (Fig. 4c; error bars on individual data points show 95% confidence intervals, revealing that the dissociations are robust). In particular, the octave, and to a lesser extent the fifth, were not rated as highly as would be predicted from their fusion (conversely, other consonant intervals are rated as more pleasant than would be expected based on their fusion, at least when compared to the fifth and octave). As a result, the correlation between mean consonance and fusion ($r_s = 0.85$) was lower than it could have been given the high reliability of the mean results ($r_s = 0.99$ and $0.98$ for fusion and consonance, respectively).

**Individual differences in fusion do not predict consonance.** The large sample size also enabled analysis of individual differences in fusion and consonance preferences. As shown in Fig. 4d, e, consonant and dissonant intervals differ on average in fusion and pleasantness, but the extent of these effects varies from person to person. If fusion directly causes consonance preferences (i.e., if listeners prefer sounds that are more fused), then individuals with large fusion effects should also have large preference effects provided the individual differences are reliable. We quantified these two effects as the difference in fusion between consonant and dissonant intervals, and the difference in pleasantness ratings between consonant and dissonant intervals, respectively. We used all the intervals in the experiment apart from the unison (consonant and dissonant interval sets were those colored blue and brown, respectively, in Fig. 4a, b).

Both the fusion and consonance effects exhibited reliable individual differences—participants with large effects on one half

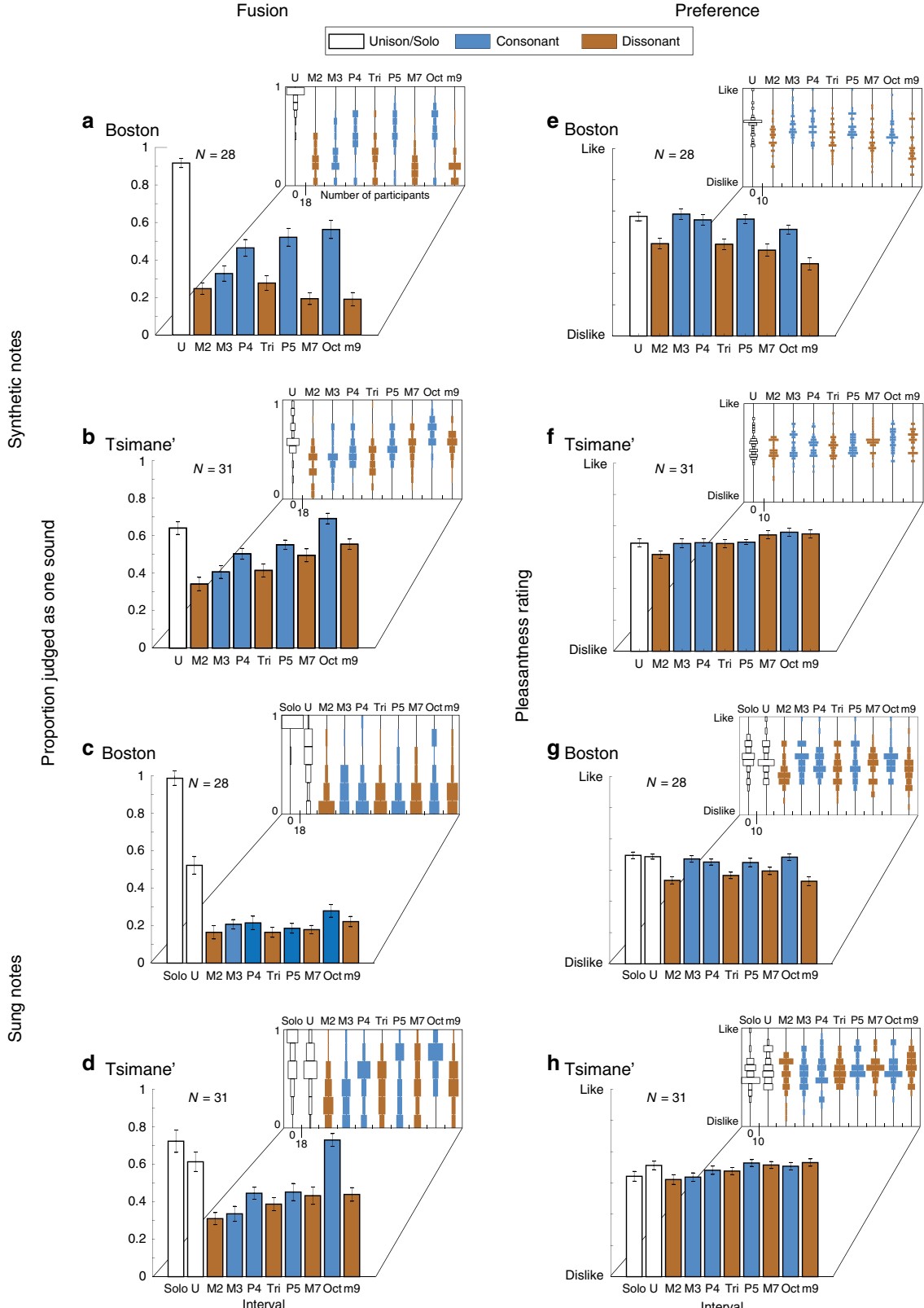

**Fig. 3 In-person fusion and preference results separated by musical interval. a–d** Proportion of stimuli judged as one sound. All bar plots show the mean ± SEM, except for panels (**d**) and (**h**) which plot the mean ± within-participant SEM. Insets show histograms of individual subject data for each condition. Due to the small number of trials per condition, and the small number of possible responses, there were a finite number of possible values for the average response for a participant. The histogram bins are matched to these values. Within each inset, the scaling of the histograms is matched across conditions, and this scaling is indicated once on the *x*-axis. **e–h** Average pleasantness ratings for each stimulus.

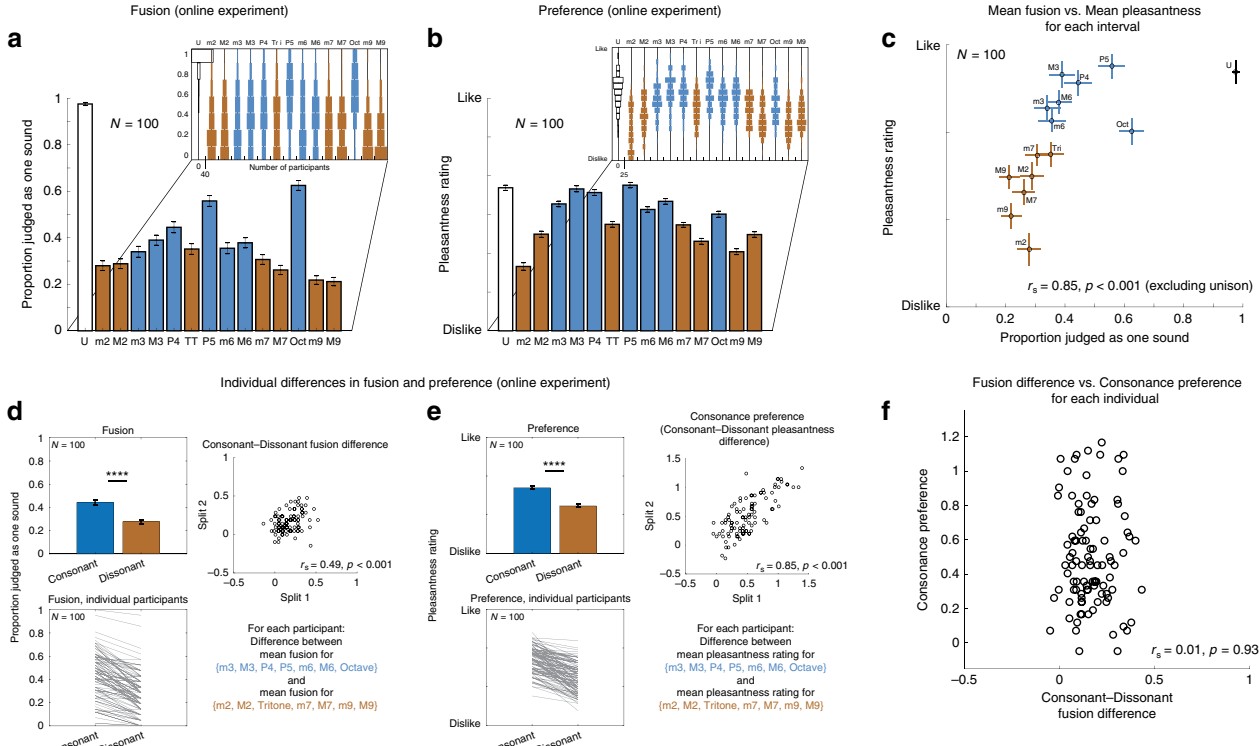

**Fig. 4 Results of online fusion and preference experiments in Western participants. a** Mean fusion of musical intervals. Here and in (**b**), bars plot mean across participants, error bars plot SEM, and insets show histograms of average responses for individual participants (same conventions as in Fig. 3). **b** Mean rated pleasantness of musical intervals. **c** Mean fusion vs. mean pleasantness for individual musical intervals (scatter plot of quantities plotted in (**a**) and (**b**)). Error bars plot 95% confidence intervals around the mean. $r_s$ value is the correlation between the mean fusion and mean pleasantness across intervals. Unison was omitted from correlation as it does not consist of two notes. **d** Individual differences in fusion. Left panels: mean fusion ± SEM for consonant and dissonant intervals, averaged across all participants (top) or for individual participants (bottom). Consonant and dissonant intervals were all those colored blue and brown, respectively, in (**a**)–(**c**). A Wilcoxon signed-rank test was used to compare the two conditions, ****$p < 0.0001$. Right panel: test–retest reliability of the difference in fusion for consonant and dissonant intervals, computed from two splits of each participant's trials. Here and in (**e**), reliabilities were Spearman–Brown corrected to best estimate reliabilities of measures derived from full experiment. **e** Individual differences in consonance preferences. Left panels: mean pleasantness ± SEM for consonant and dissonant intervals, averaged across all participants (top) or for individual participants (bottom). A two-tailed paired $t$-test was used to compare the two conditions, ****$p < 0.0001$. Right panel: test–retest reliability of the difference in pleasantness for consonant and dissonant intervals (their "consonance preference"), computed from two splits of each participant's trials. **f** Consonance preference vs. consonant–dissonant fusion difference for individual participants. Each dot represents a participant. Here and in scatter plots in (**d**) and (**e**), the $x$-coordinates of individual dots were jittered by a small amount (randomly drawn from $U[-0.01, +0.01]$) to mitigate the visual effect of dot overlap.

of the trials (split 1) also had large effects on the other half of the trials (split 2) (Fig. 4d, e, right panels; $r_s = 0.49$, $p < 0.001$ and $r_s = 0.85$, $p < 0.001$ for fusion and consonance, respectively, Spearman–Brown corrected). These reliable individual differences should be correlated if the two effects are related. However, the two effects were not correlated across participants (Fig. 4f, right panel; $r_s = 0.01$, $p = 0.93$).

The results were qualitatively unchanged if the ratings for a participant were $z$-scored prior to the calculation of their consonance preference (correlation between fusion difference and consonance preference remained non-significant; $r_s = 0.12$, $p = 0.23$). This latter analysis controls for the possibility that differences in consonance preference could be driven by differences in the use of the rating scale, which could in principle lower the correlation with the fusion effect. The results were also qualitatively unchanged if we excluded participants who were close to floor or ceiling on the fusion task, who might otherwise be expected to reduce the correlation with consonance preferences for uninteresting reasons (participants were excluded for whom mean fusion of consonant and dissonant intervals was less than 0.2 or greater than 0.8; the remaining participants yielded $r_s = 0.08$, $p = 0.47$, $N = 84$). The lack of a relationship between

fusion and consonance preference was also robust to the particular intervals included in the consonant and dissonant sets (Supplementary Fig. 3). As best we can determine, fusion and consonance preferences in Westerners are unrelated at the level of the individual listener.

Overall, these results show that even in Westerners, consonance preferences are not fully predicted by other perceptual consequences of harmonicity, either in the population average, or in individuals. Preferences are evidently subject to some other (presumably culture-specific) influence, such as experience with particular intervals in music.

## Discussion

We found that both native Amazonians and US non-musicians were more likely to fuse canonically consonant note pairs compared to dissonant note pairs, even though only listeners in the US exhibited preferences for consonance (Fig. 2). The pattern of variation in fusion across intervals was similar for both groups (Fig. 3), and was roughly what would be predicted from the similarity of consonant notes to the harmonic series: fusion was greater for the octave than for the fifth, and for the fifth than for the fourth, and for the fourth than for the third. Because Tsimane'

rarely engage in group performances and thus have limited exposure to harmony, the observed fusion of consonant intervals seems likely to result from generic mechanisms for harmonicity-based sound segregation rather than from experience with musical intervals in polyphonic music. However, although fusion was overall higher for consonant than for dissonant intervals, the pattern of consonance preferences of Western listeners (across intervals) was not predicted by that for fusion. Moreover, the extent of fusion for consonant intervals in individual listeners was unrelated to their preferences for consonance (Fig. 4). Overall, the results are consistent with the idea that the basic perceptual machinery for sound segregation is similar across cultures, producing shared representations of some musical structures. However, these shared representations do not directly determine aesthetic associations, which appear to depend critically on culture-specific musical experience.

Although the stimuli used to assess consonance preferences in this study were limited to single intervals (in order to match stimuli across the fusion and preference experiments), we have previously obtained similar preference results with extended vocal harmonies generated from excerpts of sung Tsimane' melodic phrases[31]. The differences in preferences evident across groups thus do not appear to be specific to brief stimuli. We have also previously obtained similar results with chords containing more than two notes[31]. Moreover, the Tsimane' have consistently shown strong preferences for other stimulus contrasts (here positively and negatively valenced vocalizations, and smooth and rough synthetic tones), indicating task comprehension and compliance. The group differences in consonance preferences appear to reflect differences in the stimulus variables that drive aesthetic responses, with Western notions of consonance not influencing Tsimane' notions of pleasantness.

The increased fusion we found for consonant compared to dissonant intervals is consistent with previous studies of fusion in Westerners[18,33,35], but utilizes modern psychophysical methods and large samples, tests both synthetic and sung notes, and examines a broad range of intervals. In addition, we provide the first evidence that the phenomenon of fusion is present cross-culturally. Previous findings that non-human animals can discriminate consonant from dissonant chords[44,45] have raised the possibility that the distinction might be intrinsic to the auditory system, but the paradigms involved required extensive training. Moreover, recent evidence for species differences in cochlear frequency selectivity[46] and in the processing of harmonic sounds[47–50] suggest that perceptual equivalences related to harmonicity are likely to differ in at least some respects between humans and nonhuman animals. Cross-cultural experiments in humans are thus essential to testing the universality of music perception. We found cross-cultural similarity in fusion using a paradigm that did not require training, indicating that at least for humans, musically relevant properties of harmonicity-based sound segregation are a widespread property of human audition.

Our findings could help explain previous findings in Western infants, who have in some cases been reported to produce longer gaze durations for consonant compared to dissonant intervals[51,52] (though not always[53], and though some such results were plausibly driven by roughness[51], which we find to be aversive even to the Tsimane', and which does not predict consonance preferences in Westerners[24,26,29]). These looking-time effects have been interpreted as an indication of innate preferences, but could reflect any perceptual difference between the stimuli. Our results suggest that fusion differences between consonant and dissonant intervals are present independent of musical experience, such that infants might also hear consonant intervals as more similar to a single sound, potentially drawing their attention and explaining prior results without the need to posit an affective response. This idea is consistent with prior evidence that infants treat distinct consonant intervals as perceptually similar to each other[54].

Our results here contrast with other recent experiments in the Tsimane' probing another aspect of harmonic sounds—their pitch. Whereas US participants tend to reproduce heard melodies using sung notes transposed by an integer number of octaves, and to match the pitch of melodies in their singing range, Tsimane' do not. Tsimane instead reproduce primarily the relative pitch relations between notes[55]. The consequence of this is that even though the Tsimane' fuse concurrent notes separated by an octave, they do not seem to hear those two notes as similar when they are presented separately, unlike Westerners. Together, the results are consistent with the idea that harmonicity-based sound segregation and musical pitch are partially dissociable[56,57]. In particular, the results raise the possibility that the basic properties of pitch perception depend more strongly on particular types of musical experience than do the basic properties of harmonicity-based sound segregation.

Musical phenomena that are present independent of musical experience are of interest in part because they are natural candidates to constrain musical systems. Although the Tsimane' do not engage much in group singing, our results suggest that if they tried to do so, the musical system they might converge on would likely be constrained by the perceptual effects of fusion that we observed here. The fusion of simple-integer-ratio pitch intervals could make such intervals easier to produce in group settings, for instance, by providing a signal when the interval is correctly produced. Fusion could thus potentially bias musical systems towards the use of simple-integer ratios, in particular the octave, which exhibited relatively strong fusion across participant groups and experiments. In addition, fusion differences between consonant and dissonant pitch combinations may create a natural contrast to exploit in musical systems that use harmony[58].

The picture suggested by our experiments is that universal perceptual biases exist that could partially constrain musical systems, but that they only indirectly shape aesthetic responses, which are instead determined by what is prevalent in the musical systems an individual experiences. This general picture is reinforced by the results in US listeners, in whom the average pattern of fusion and consonance judgments was distinct at the level of specific intervals, and in whom the increase in fusion for consonant intervals did not predict preferences for consonant intervals within individual listeners. Fusion seems plausibly to have had some influence on Western music, in that fusion distinguishes intervals prominent in Western music (and in the music of some other cultures) as perceptually privileged. Indeed, consonance (here operationalized as pleasantness) and fusion are partially correlated across intervals. But consonance appears to be influenced by additional factors—it is not simply the result of positive valence applied to the intervals that exhibit high fusion.

Previous results have linked consonance and harmonicity via correlations between preferences for consonance and preferences for harmonicity[24,26,29]. The present results indicate that these correlations are likely not driven by variation in harmonicity-based sound segregation. One possibility is that the pleasantness of musical intervals is driven by their familiarity, and that other harmonic sounds become pleasant by association because the intervals prevalent in Western music tend to be harmonic. The present results motivate revisiting the link between harmonicity and interval preferences, and between these preferences and other perceptual phenomena related to harmonicity, such as f0-based pitch.

Pitch intervals can be produced melodically (sequentially) as well as harmonically (concurrently), raising the question of whether melodic intervals defined by simple integer ratios would also in some way exhibit a privileged perceptual status[59] cross-

culturally. We focused on intervals with concurrent notes because there is (1) a plausible mechanism to distinguish them (harmonicity-based sound segregation), (2) an associated psychophysical task that is robust and easily described (fusion), and that can thus be conducted on diverse participant groups, and (3) a well-documented aesthetic response in Westerners (none of which are available at present for melodic intervals). Although it remains to be seen whether some melodic intervals might be analogously distinguished, our results support the idea that perceptual effects related to simple-integer ratios in musical intervals could drive cross-cultural regularities in musical scales[1–3,60], just as the perceptual prominence of temporal patterns related by simple integer ratios may constrain musical rhythm[61,62]. We note that factors independent of perception, such as the ease of consistently producing pitch intervals with simple integer ratios on musical instruments (e.g., by bisecting strings), could also contribute to such regularities, and that the imprecision of sung intervals[63] must also be reconciled with such accounts.

We also note that the presence of constraints that bias musical systems in particular directions does not preclude cross-cultural variation[64]. For instance, vocal harmony with intervals considered by present-day Westerners to be dissonant, such as the major second, is thought to have been relatively common in other cultures and historical periods[65]. Biases imposed by perceptual fusion may have contributed to statistical regularities in the pitch intervals used across cultures, but are clearly biases rather than absolute constraints.

Our experiments with the Tsimane' illustrate the opportunities provided by cross-cultural research to dissociate perceptual effects that are widespread in Western listeners. Scientists have long noted that generic mechanisms for representing harmonic sounds could explain aspects of music perception, but most relevant experiments have been limited to Western participants, leaving the universality of any such effects unclear. Here we show cross-cultural consistency in the representation of musical structure related to the harmonic series despite marked differences in the aesthetic associations for that structure. The results suggest universal features of perception that may shape musical behavior around the world, but also indicate the rich interplay with cultural influences that give rise to the experience of music.

## Methods

**Participants—Boston**. 28 Participants (14 female) were tested (mean age = 33.7 years, S.D. = 8.9 years). All participants reported no formal musical training. On average, participants had 15.6 years of schooling (S.D. = 2.7).

A separate group of 14 Boston participants completed the fusion control experiment with harmonic and inharmonic concurrent voices (Fig. 2c, 8 female, mean age = 31.9 years, S.D. = 8.8 years). All 14 reported no formal musical training. On average, participants had 14.6 years of schooling (S.D. = 2.1 years).

**Participants—Tsimane'**. 32 Participants (14 female) from the villages of Mara, Moseruna, Emeya, and Donoy completed the main fusion and preference experiments. One was removed from the subsequent analysis because they appeared to reverse the instructions. The remaining 31 participants (14 female) had a mean age of 23.4 years, S.D. = 5.7 years. Participants had an average of 3.5 years of schooling (S.D. = 2.7). Tsimane' schooling was not directly comparable to schooling in the US, being more intermittent and with a variable curriculum, dependent on individual teachers. A subset (N = 23) of these 31 participants completed the first fusion control experiment (Fig. 2b). An additional 10 participants began the experiments but did not complete them for various reasons (e.g., restless or sick children who needed attention, or noncompliance with instructions), and their data were not analyzed. Tsimane' participants generally had little musical experience, and none regularly played a musical instrument.

A separate group of 21 Tsimane' participants (from the villages of Mara and Emeya) completed the fusion control experiment with harmonic and inharmonic concurrent voices (Fig. 2c, 5 female, mean age = 26.8 years, S.D. = 8.9 years). Participants had an average of 2.4 years of schooling (S.D. = 2.3).

The Tsimane' are an indigenous people of lowland Bolivia, who live in small villages along the branches of the Maniqui and Apere rivers (Fig. 1b). They subsist mostly on horticulture, hunting, fishing, farming, and plant collection. Over the course of our recent trips to the Tsimane' territory, it has become apparent that the

region is undergoing rapid modernization, with changes evident even year to year due to a push by the Bolivian government to provide modern services to the indigenous peoples[43]. Some villages now have electricity (when we first visited the region in 2011 this was not the case). Evangelism has also spread to many villages through Christian missionaries. Until recently, radios were rare in Tsimane' villages due to the limited availability of batteries, but their usage has increased across the Bolivian lowlands in recent years. For this paper we thus ran our experiments in some of the relatively isolated and non-Westernized villages that remain.

We tested participants in four Tsimane' villages. Two villages (Mara and Moseruna) were a 2-day walk or a 3-hour car ride from San Borja, along a road that was only accessible to high clearance vehicles and motorcycles if recent weather had been dry. The other villages (Emeya and Donoy) were located along the Maniqui river and only accessible by a 2–3-day trip on a motorized canoe. Emeya and Donoy are among the most remote Tsimane' villages.

**Sample sizes for in-person experiments**. Sample size for Boston was chosen based on an a priori power analysis of pilot fusion judgment data with synthetic note intervals. We measured the split-half reliability of the pattern of fusion across all intervals, averaged across different numbers of participants. We chose a sample size expected to yield a split-half reliability exceeding $r = 0.95$, 90% of the time (18 participants). The sample size for the Tsimane' group was as large as possible given practical constraints and was larger than the Boston group.

**Online participants**. 147 Participants completed the online experiments. We used a geographic filter to restrict participation to individuals with IP addresses in the United States or Canada. 47 were removed for not performing perfectly on the control task (though results of the main experiments were similar in these participants). The results reported here reflect data from the remaining 100 participants (43 female, mean age = 38.2 years, S.D. = 10.3 years). All participants reported no formal musical training.

**Sample sizes for online experiments**. We aimed to collect at least 92 participants based on an a priori power analysis; this sample size was chosen so that we could detect a correlation of $r = 0.3$ between fusion and preference measures from the two experiments with a significance threshold of $p < 0.05$, 90% of the time.

**Background information on Tsimane' music**. The Tsimane' have traditional indigenous music, familiarity with which varies across individuals. To the best of our knowledge, traditional songs were sung by individuals one at a time, generally without instrumental accompaniment[41]. The most common type of musical instrument we have encountered is a flute, but other string instruments and drums are also common. As reported by Riester[42], Tsimane' song melodies are characterized by the use of a small number of musical pitches forming small intervals, often approximately two or three semitones, and a narrow vocal range. In addition to their knowledge of traditional music, nowadays many Tsimane' villagers are somewhat familiar with religious Christian hymns, which they learn from missionaries. These hymns are monophonic and sung in Tsimane'. They are similar to traditional Tsimane' music in that they rely on small intervals and a narrow vocal range. Group singing appears to be rare, irrespective of whether the material is traditional songs or hymns. Tsimane' participants thus may have had prior familiarity with melodic versions of some of the intervals used in the present experiments, but likely much less so with the harmonic versions that were actually used. Most Tsimane' report that they do not regularly play musical instruments.

Our knowledge of Tsimane' music derives in part from a series of interviews and recording sessions that we conducted in the summer of 2015, as part of a previous study[31]. We set up a makeshift recording studio in the town of San Borja, and over the course of 4 days spoke with 10 Tsimane' musicians about their musical practices. We also recorded performances of their songs. Since then we have continued to learn about their musical experience during visits to their villages.

**Stimulus presentation for in-person experiments**. Stimuli were played by MacBook Air laptop computers using over-ear closed headphones (Sennheiser HD 280Pro), at a presentation level of 70 dB SPL. These headphones are designed to attenuate ambient noise (by up to 32 dB depending on the frequency) and are thus well suited for experiments in outdoor or public settings. Presentation was diotic with the exception of the roughness experiment (see below). The audio presentation system was calibrated ahead of time with a GRAS 43AG Ear & Cheek Simulator connected to a Svantek's SVAN 977 audiometer. This setup is intended to replicate the acoustic effects of the ear, measuring the sound level expected to be produced at the eardrum of a human listener, enabling sound presentation at the desired sound pressure level.

**Experimental protocol for in-person experiments**. All experiments were completed in a single session for each participant, ranging from 30 to 60 min in duration. Preference and fusion experiments were conducted in separate blocks and the order of the two blocks was randomized across participants in each experimental group. Within the preference block, the order of experiments was

randomized. The three blocks of fusion trials (synthetic notes in equal temperament tuning, synthetic notes in just intonation tuning, and sung notes) were also completed in a random order. The first fusion control experiment (Fig. 2b) was also randomly positioned within this set for the 23 participants who completed it.

After the participant heard each stimulus, they gave a verbal response ("like it a lot", "like it a little", "dislike it a little", or "dislike it a lot" for preference tasks and "one" or "two" for fusion tasks). Instructions and responses were in English in Boston, and in Tsimane' for all Tsimane' participants. For Tsimane' participants, translators (who spoke Tsimane' and Spanish) delivered the instructions and interpreted the participants' verbal responses. Experimenters were trained to recognize the Tsimane' words for the response options so they could evaluate the correctness of the translator's response. The experimenter entered the response into a MATLAB interface. For both tasks, the translation of the instructions was checked by having one translator produce an initial translation, and then asking a second translator to translate the Tsimane' instructions back into Spanish.

We operationalized consonance as pleasantness because lay listeners might not be familiar with the term "consonance", and because it was impossible to translate into Tsimane'. We are cognizant that consonance is a multifaceted phenomenon in the context of music theory[15,36]. Here we assessed pleasantness by querying "liking" for ease of translation. We have found previously that the liking task with four response choices (described above) yields similar results to a pleasantness rating scale task[24,31].

Experimenters were blind to the stimuli being presented to avoid biasing responses or data entry. Stimuli were only audible to the participant (the sound attenuation from the closed headphones coupled with the distance between experimenter and participant and the background noise from wind and other environmental sounds was sufficient to achieve this). There was no feedback provided to participants. Experimenters were likewise unable to see whether a response was correct or incorrect (for the fusion experiments in which there was a correct response on each trial).

Experiments were conducted in August 2019 (Tsimane') and September–November 2019 (USA). In-person experiments were conducted in Tsimane' villages (for Tsimane' participants) and in public spaces on or near the MIT campus (for Boston participants).

Because Tsimane' villages lack enclosed or sound-proof spaces, we optimized listening conditions by reducing noise in the experimental area, selecting locations that were as distant as possible from potential acoustic disturbances from community activities, and typically assigning a team member to keep children and animals out of earshot from the experimental stations. Additionally, to help ensure that differences in testing conditions between groups did not influence findings, we conducted experimental sessions for Boston participants in public areas of the MIT campus and surrounding neighborhood. We chose locations with consistent low background noise from students and staff walking by or studying at nearby tables (e.g., the student center and a public atrium), with the intention of matching conditions to those in the field as best possible. Our subjective impression is that the Boston testing locations had somewhat more distractions and background noise than our testing locations in Bolivia. The responses we measured in Boston were not obviously different from those of similar experiments we have previously conducted in soundproof booths, indicating that results of this sort are fairly robust to the testing conditions. Nonetheless, to minimize the audibility of the background noise in the different experimental settings, we used closed circumaural headphones. The same closed circumaural headphones and computers were used with both groups.

The study was approved by the Tsimane' Council (the governing body of the Tsimane' in the Maniqui basin, where the experiments took place), and the Committee on the Use of Humans as Experimental Subjects at MIT. Experiments were conducted with the informed consent of the participants, and informed consent was obtained to publish the image in Fig. 1c. Participants were compensated for their time with money (US and online participants), or with culturally-appropriate packages of goods (Tsimane' participants).

**Stimuli—concurrent talkers**. Stimuli for the first control experiment were generated by excising approximately 1 s of continuous speech (usually a single word, sometimes a short phrase with no break in the acoustic waveform), from recordings of Tsimane' speakers. The recordings were made in a different village from those in which the experiment was run. We generated eight trials with a single female speaker, eight trials with a single male speaker, and 16 trials where utterances from a male and a female speaker were presented concurrently. All audio was gated with 15 ms half-Hanning windows to avoid onset transients.

**Stimuli—harmonic and inharmonic sung vowels**. Stimuli for the second control experiment were similar to those in the musical interval experiments, except that the f0 difference between the vowels was selected to minimize the chances of fusion, and the sung vowels were resynthesized to be either harmonic or inharmonic[9,66]. Vowels were excerpted from recordings of R&B singers singing a chromatic scale taken from the RWC Instrument Database[67] (also used for the sung interval stimuli described below). For the inharmonic condition, the frequency of each frequency component, excluding the fundamental, was jittered by an amount chosen randomly from a uniform distribution $U(-0.5, 0.5)$. The sampled jitter values were multiplied by the f0 of the vowel and added to respective

harmonic frequencies[13]. To minimize differences in beating between conditions, jitter values were selected via rejection sampling such that adjacent frequency components were always separated by at least 30 Hz. In the one-voice conditions, the vowel recording was randomly chosen to be 1–2 semitones either below or above 200 Hz. Vowels "A", "I", "U", or "O" (cardinal vowels) were used with equal probability. Half of the singers were male and half were female. In the two-voice conditions, the lower voice was randomly chosen to be 1–2 semitones either below or above 200 Hz. We used vowel pairs that were adjacent in vowel space: ["A", "I"], ["I", "U"], ["U", "O"], and ["O", "A"]. The lower voice was that of a male singer, and the upper voice was that of a female singer. The interval between the two vowels was either a minor third, tritone, minor sixth, or minor seventh, in equal temperament. Our pilot experiments indicated that none of these intervals would produce substantial fusion in Western participants. The goal of the experiment was to assess the dependence of concurrent source segregation on harmonicity in conditions where segregation would normally occur for harmonic sources.

Because the singers were not perfectly in tune, we imposed minor pitch-shifts on the sung audio, adjusting the mean pitch of each audio excerpt to exactly match the intended intervals. The f0 contour was otherwise unmodified, such that the stimuli retained the small f0 fluctuations present in the original recordings. Stimuli were generated from the first 500 ms of the voiced component of each vowel, conservatively estimated as starting 20 ms after the onset of voicing as determined by STRAIGHT[68]. The resulting audio was gated with 15 ms half-Hanning windows. All manipulations were performed using STRAIGHT[68].

For the 17 participants run in the village of Mara, the experiment had 64 trials (16 trials per condition, harmonic/inharmonic crossed with one-voice and two-voices; individual vowels or vowel combinations were counterbalanced across conditions, and F0s were randomized). Results were sufficiently robust that for the final 4 participants, run in Emeya, the experiment was cut in half to save time. For both groups we pre-generated ten different sets of stimuli with random pairings of specific singer, root pitches, and vowels; each participant was randomly assigned one of these sets of stimuli during the experiment.

**Stimuli—vocalizations**. Vocalization stimuli were identical to those from a previous study[31]. Vocalizations were a subset of the Montreal Affective Vocalization Set[69], which consists of recorded vocalizations produced by actors in a laboratory setting. Five vocalizations were selected from each of the categories of laughter, gasps, and crying, which had previously been found to be rated as pleasant (laughter) and unpleasant (gasps and cries) by both US[26] and Tsimane' listeners[31]. The stimuli were presented in random order in a single block of 15 trials. The crying sounds gave similar results to the gasps (low ratings by both groups) and their ratings were omitted from the analysis to simplify the data presentation to two stimulus conditions per experiment.

**Stimuli—smooth and rough tones**. The smooth and rough tone stimuli were generated as in previous studies[24,31], by presenting pairs of single frequencies to either the same or different ears (diotic and dichotic presentation, respectively). Diotic presentation of two similar but non-identical frequencies produces the "rough" sensation of beats, typically considered unpleasant by Western listeners. In contrast, dichotic presentation of two such frequencies greatly attenuates perceived beats, but leaves the spectrum unchanged relative to the diotic version. Stimuli were generated in three different frequency ranges. The frequencies composing each stimulus were separated by either 0.75 or 1.5 semitones (1.5 for the low- and mid-frequency ranges, and 0.75 for the high-frequency range, to produce beat frequencies with prominent roughness), such that considerable beating was heard when presented diotically. The lower of the two frequencies was set to 262, 524, or 1048 Hz, plus or minus an offset of one or three semitones. Each stimulus was presented four times. This resulted in 2 conditions × 3 frequency ranges × 4 frequencies = 24 trials presented in a single block in random order. The block also included 4 trials with pure tone stimuli, the results of which are not analyzed here.

**Stimuli–synthetic intervals for fusion and preference experiments**. Stimuli were composed of two notes. Each note was a synthetic tone that was 2000 ms in duration and contained harmonics 1–12 (in sine phase). To mimic the acoustics of many musical instruments, harmonic amplitudes were attenuated by −14 dB/octave, and tones had temporal envelopes that were the product of a half-Hanning window (10 ms) at either end of the tone and a decaying exponential (decay constant of 4 s$^{-1}$).

The two notes were presented concurrently. The pitch interval between the two notes was either a unison, major second, major third, perfect fourth, tritone, perfect fifth, major seventh, octave, or minor ninth. These intervals were chosen to include the four most canonically consonant intervals (the octave, fifth, fourth, and third) along with a selection of canonically dissonant intervals, chosen to approximately alternate with the consonant intervals when ordered according to the interval size in semitones. Interval size was thus dissociated from similarity to the harmonic series (i.e., Western consonance, associated with simple integer ratios).

Interval stimuli were presented in three different pitch ranges, with root note fundamental frequencies either zero or two semitones above 110, 220, and 440 Hz. Each such stimulus was presented once, resulting in 54 trials (9 intervals × 3 f0 ranges × 2 root f0s) for each preference experiment, presented in random order.

There were two versions of each experiment, one with intervals in equal temperament and one with just intonation (see Supplementary Table 1 for interval ratios/sizes). The two versions of each experiment were conducted in random order within the larger sets of experiments as described above.

**Stimuli–sung intervals for fusion and preference experiments.** Stimuli for the sung note experiment were generated from recordings of R&B singers taken from the RWC Instrument Database[67]. R&B singers were chosen instead of opera singers in the set because they used less vibrato. The lower note was always chosen from a male singer, and the f0 of this root note was selected to be either 200 Hz, or 1–2 semitones below or above 200 Hz (yielding 5 root pitches). The second note (for all intervals, including the unison) was chosen from a female singer's recordings (these recordings were made in semitone steps, such that there was always a recording close to the desired f0). We tested the same 9 intervals as in the synthetic-note version of the experiment, plus an additional condition with only one male singer ("Solo"). Each interval was presented five times, each time with both singers singing the same vowel, either "A", "E", "I", "O", or "U", and each time using one of the five possible root pitches. We pre-generated ten different sets of stimuli with random pairings of singers, root pitches, and vowels; each participant was randomly assigned one of these stimulus sets during the experiment. The experiment thus contained 50 trials (10 intervals × 5 exemplars varying in vowel and root f0), presented in random order.

Because the singers were not perfectly in tune, we imposed minor pitch-shifts on the sung audio, adjusting the mean pitch of each audio excerpt to exactly match the intended intervals (in just intonation; equal temperament was not used for this experiment). Such shifts were always less than half a semitone. Additionally, we standardized the pitch modulation (vibrato) across notes. We modified the pitch contour of each note so that the standard deviation was 1% of the mean f0. We performed these manipulations using STRAIGHT[68]. The notes used in the stimuli were the first 500 ms of the voiced component of each vowel, conservatively estimated as starting 20 ms after the onset of voicing as determined by STRAIGHT. The resulting audio was gated with 15 ms half-Hanning windows.

**Experimental protocol for online experiments.** Online participants were recruited through Amazon's Mechanical Turk platform, using a geographic filter to restrict participation to individuals logging on from the United States or Canada. Participants began by completing a demographic survey followed by a brief "headphone check" experiment intended to help ensure that they were wearing headphones or earphones[70]. Participants who passed the headphone check then completed the first control experiment from the in-lab experiment (one vs. two concurrent talkers), followed by one experiment measuring fusion and one measuring pleasantness. The order of these latter two experiments was randomized across participants. The instructions and response choices were identical to the experiments run in-person. Stimuli for the fusion and pleasantness experiments were identical to those used for in-person synthetic interval experiments (three different pitch ranges, with root note fundamental frequencies either zero or two semitones above 110, 220, and 440 Hz), but with intervals ranging from 0 to 14 semitones in integer steps (we only tested intervals in equal temperament for these experiments).

We included for analysis all participants with 0 years of self-reported musical experience who performed at ceiling on the control experiment (which here was intended to identify noncompliant or inattentive online participants). We have previously found that data quality from online participants can be comparable to that in the lab provided steps such as these are taken to ensure task compliance[37].

**Analysis.** Because there was no detectable effect of tuning system (see "Results" section), in the main analyses of the fusion and preference experiments with synthetic notes (Fig. 2f, h), the proportion of trials judged to be one sound for each participant, or the mean rating of a sound for each participant, were collapsed across tuning system and averaged across the trials for a given interval. For Fig. 2f–i and most further analyses, results were then averaged across consonant intervals (major third, perfect fourth, perfect fifth, and octave) and dissonant intervals (major second, tritone, major seventh, and minor ninth), and statistics were performed on these mean ratings for each participant. For other preference experiments, the ratings for each participant were similarly averaged across the exemplars for a condition.

For analyses of online experiments (that included additional intervals), the minor third, minor sixth, and major sixth were classified as consonant, and the minor second, minor seventh, major ninth were classified as dissonant. For analyses of individual differences, the consonant–dissonant fusion difference was computed for each participant by averaging the results for each interval within the consonant set and subtracting the average of the results for the dissonant set. The consonance preference was analogously computed for each participant by averaging the mean pleasantness rating for each interval within the consonant set and subtracting the mean rating for the dissonant set. The reliability of these effects was computed as the correlation between the effects measured using two splits of the trials for each condition of the experiment. These splits were chosen to balance the stimulus register, with one split containing the lower of the two root-note f0s used in a register, and the other split containing the higher of the two f0s. The

results of Fig. 4f were robust to the exact choice of intervals in the consonant and dissonant sets (Supplementary Fig. 3).

**Statistics and reproducibility.** Data distributions were evaluated for normality by visual inspection. The data from the fusion experiments were often non-normal due to the prevalence of response proportions near 0 and 1, and so we relied on non-parametric tests. Significance of pairwise differences between fusion conditions was evaluated using a two-sided Wilcoxon signed-rank test. For multi-condition tests of fusion judgments (analogous to ANOVAs), we computed $F$ statistics and evaluated their significance with approximate permutation tests, randomizing the assignment of the data points across the conditions being tested, 10,000 times.

For the preference experiments, two-tailed paired $t$-tests were used to test for differences between conditions within groups (Cohen's $d$ was used to quantify effect sizes), and mixed-design ANOVAs were used to test for main effects of stimulus condition and for interactions between the effect of stimulus condition and participant group. Mauchly's test was used to test for violations of the sphericity assumption. The test never indicated violations of sphericity, so it was unnecessary to use a correction. For multi-condition tests on results of the sung note experiments (both fusion and pleasantness ratings), the solo condition was omitted to match conditions with synthetic note experiment (e.g., to test for interactions between interval and experiment).

Error bars on plots are standard error of the mean, except for the graphs showing results for sung notes (Figs. 2g, i and 3d, h) and mean fusion/pleasantness of individual intervals from the online experiment (Fig. 4d). Within-participant standard error of the mean (SEM) was used for the results with sung notes because there were large differences between individual participants in both groups in the overall tendency to report hearing one sound, potentially due to the fact that each participant heard one of 10 different stimulus sets, that otherwise masked the consistent differences between consonant and dissonant intervals. 95% confidence intervals were used in Fig. 4d to facilitate comparison of individual data points. Spearman (rank) correlations were used to examine the relationship between fusion ratings and preference ratings for online experiments; split-half reliabilities of the fusion and consonance measures were Spearman–Brown corrected to obtain the estimated reliability of the measures for the full data set.

All in-person experiments measuring fusion with synthetic notes were replicated once with similar results in the Tsimane' (using slightly different task instructions and musical intervals), and twice with similar results in Boston participants (again with slightly different task instructions and musical intervals). In-person experiments measuring pleasantness in the Tsimane' have been replicated twice with slightly different sets of musical intervals, obtaining similar results (one replication was previously published[31]).

**Reporting summary.** Further information on research design is available in the Nature Research Reporting Summary linked to this article.

## Data availability

All data are provided as a Source Data file in the Supplementary Information. The RWC Instrument Database can be downloaded at https://staff.aist.go.jp/m.goto/RWC-MDB/. The recorded vocalization stimuli used in this study are available from the authors upon request. A reporting summary for this Article is available as a Supplementary Information file.

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

## Acknowledgements
The authors thank T. Huanca and E. Conde for logistical support in Bolivia; translators D. Nate Anez, S. Hiza Nate, R. Nate Roca, M. Roca Moye; canoe motorist I. Canchi Bie; drivers H. Roca Velarde and P. Roca Ramirez; R. Gonzalez for logistical support in Boston; E. Pollack for collecting pilot data; R. Grace for assistance with Boston data collection; and N. Kanwisher, M. Cusimano, M. Weiss, and the entire McDermott Lab for comments on the manuscript. Work supported by NIH Grants F31DCO18433 and R01DC014739, and an NSF Graduate Research Fellowship to M.J.M.

## Author contributions
M.J.M. and J.H.M. designed the experiments. All authors collected data. M.J.M. and J.H.M. analyzed the data and drafted the manuscript. M.J.M. made the figures. All authors edited the manuscript.

## Competing interests
The authors declare no competing interests.
