## [Peer Review File · Nature Communications]

Reviewers' comments:

Reviewer #1 (Remarks to the Author):

This manuscript describes the results of a study that follows up on the earlier study of McDermott et al. (2016; *Nature*, 535, 547-550). The earlier study demonstrated that Western concepts of musical consonance and dissonance were not universal by showing that members of the Tsimane in a remote region of Bolivia did not express a preference for consonant over dissonant musical intervals. This interesting finding was bolstered by showing that the Tsimane were able to discriminate harmonic from inharmonic stimuli – they just didn't "prefer" the harmonic over the inharmonic, in contrast to the judgments of Western listeners.

There is much positive about this new contribution. It is well written and provides a very clean replication of the original findings concerning a lack of preference for consonant over dissonant musical intervals. This is reassuring and helps strengthen the original claims of the authors. The main claim to novelty of the current submission is the finding that the Tsimane can distinguish to some extent dissonant from consonant intervals. This is demonstrated by showing that both Tsimane and Westerners (Boston non-musicians) are more likely to rate consonant than dissonant intervals as being one sound, rather than two. Although consistent with earlier findings, this new study does not provide much that is novel or scientifically significant.

The main finding, that Tsimane can distinguish harmonic from inharmonic sounds, was already reported in the original 2016 study (their Fig. 4). The difference here is that musical intervals, rather than artificially mistuned tones, are used, and that the judgement is "one tone" vs. "two tones", notionally targeting the concept of fusion. However, all the listeners were first trained with feedback on what "one tone" and "two tones" are supposed to sound like, and the training stimuli were not all that different from the ones used in the main experiment. As shown by the authors, the "consonant" intervals used (octave, fifth, fourth) are all good approximations to a single harmonic series, with (in the case of fifth and fourth) just a few missing harmonics. Because the tones were synthesized to have identical onsets and temporal envelopes, there is not much in the way of acoustic cues to indicate that there are actually two sounds present. In contrast, the "dissonant" intervals clearly do not follow a single harmonic series and are in that way much more like the inharmonic sound used in training (which is, in fact, a musical minor 9th – one of the stimuli used in the main experiment). Thus, it is not clear how the work extends beyond the original conclusion that the Tsimane (and by extension humans in general) can distinguish harmonic from inharmonic sounds. Indeed, as it has previously been shown that this ability extends to other species (e.g., Izumi, 2000, *JASA*; Hulse et al., 1995, *JEP:General*), it would be particularly surprising if the Tsimane were unable to perform this distinction with training. The fact that over 60% of the remote Tsimane needed to repeat the training/screening session because they initially scored under 60% correct suggests that the distinction based on perceived fusion was not particularly natural, even if they could eventually perform it.

This problem also relates to the title of the manuscript: The ability to distinguish harmonic from

inharmonic stimuli does not necessarily imply anything about consonance and dissonance, which by definition are subjective labels that have already been shown not to be universal. It is therefore far from clear what the authors mean to imply by claiming “universal representations of consonance and dissonance.”

Other than this crucial major concern, there are some minor issues that should be addressed, as follows:

Line 47. The references to papers on detection of mistuned harmonics are unfortunate here, as there are counter-examples (e.g., Roberts and Brunstrom, 1998, JASA) showing that components mistuned from a regularly spaced but inharmonic complex tone can also be heard out.

Line 64. Pick f_0 or F_0 and then stick to it.

Lines 65-67. I don't know what remains “unclear” here. The results of McDermott et al. (2016, Nature) are reasonably clear on this question: inharmonicity can be detected by non-Western listeners, but it isn't necessarily judged as more unpleasant than harmonicity.

Line 126. None of the cited references mistuned all the even from the odd harmonics. There are papers that have done this, but the cited references only mistuned a single component.

Lines 130-134. This is problematic experimentally. The only sounds that could result in feedback telling participants that they were incorrect were the inharmonic sounds, if the participants responded “one sound”. Assuming that listeners can distinguish harmonic from inharmonic (as demonstrated by McDermott et al., 2016), it seems reasonable that listeners would quickly learn to respond “two sounds” to the inharmonic stimuli based on the training feedback. Although the claim that harmonic sounds are more “fused” is very reasonable, the paradigm is somewhat flawed and in danger of being circular.

Line 339. This statement is in danger of being perceived as severely disingenuous. The 2014 paper cited here throws serious doubt on earlier claims that infants look longer (or “prefer”) consonant over dissonant intervals, and yet it is cited here as if it is one of the earlier studies that reports just such a finding.

Lines 412-413. It is not necessary to report mean age or years of schooling to 4 significant figures.

Line 466. Given the discrete nature of many of the variables, wouldn't a Spearman's rank-order correlation be more appropriate?

Supplementary Figure 3a. Typo in heading (judgments).

Reviewer #2 (Remarks to the Author):

The work examines the ability of listeners naïve to Western music to distinguish two-note chords based on major or minor intervals from single sounds. The work represent an almost unique examination of listeners without Western music exposure who will likely become rarer based on the shifting demographic of this group. The more Western listeners were better at detecting the presence of two-note chords overall. But the key finding is that the naïve listeners demonstrated the same effect of whether the note combinations were consonant or dissonant on detecting note pairs despite the fact that they did not recognise these intervals.

The data support a difference in the fundamental encoding of consonant and dissonant chords which is plausibly explained by similar sensitivity to the harmonicity of complex sounds in the different groups despite the different encultured experience of major and minor keys. I am not surprised by this, considered from an auditory neuroscience perspective: the two- versus one-note distinction can be explained based on harmonicity which is a strong behavioural grouping cue and for which hard-wired neurophysiological bases have been demonstrated in animal models. But considered from a musical perspective it is important new information about the basis for learned consonance versus dissonance which cannot be attributed to differences in fundamental sensory representation and grouping.

Reviewer #3 (Remarks to the Author):

The manuscript reports a set of behavioral experiments that compared musical interval perception in Western listeners and in Tsimane' listeners. The aim is a cross-cultural comparison of the perception of 'fusion' and 'consonance' for listeners with a varying degree of familiarity with Western music. Two main tasks were run, in both groups: a fusion task, for which listeners had to report whether they heard a single tone or two tones for various intervals, and a consonance task, for which listeners rated the pleasantness of these same intervals. All groups of listeners showed a greater tendency to report a single sound, so more fusion, for consonant intervals. However, there was no preference for consonant intervals in Tsimane' listeners, unlike for Western listeners and even to some extent for Westernized Tsimane' listeners.

This study is a new installment in a rich series of experiments comparing Tsimane' and Western listeners for various perceptual task, using musical or music-related material. As with previous papers, the question asked is relevant and interesting, the experimental procedures are impressive and solid, and the dataset is highly valuable and unique. An added twist is that there are now correlations found with the degree of Westernization of Tsimane' listeners, underscoring the timeliness of such comparative studies. Finally, the manuscript is very well written and the results clear-cut. As such, I have very little remarks for improvements directly related to my core field of expertise.

The only set of questions I would like to raise is about the link between fusion and consonance. Fusion is a concept from experimental psychology, whereas of course consonance has a long musical history. As

underlined in the manuscript, a link between the two has been suggested 150 years ago by Stumpf. But, since then, such a link has been debated (for instance, see the recent review by Parncutt and Hair, *Journal of Interdisciplinary Music Studies*, 2011). Reading the manuscript, I am not fully sure whether the authors have shown a common basis for consonance judgments, or rather brought strong contradictory evidence against Stumpf's proposal.

General comments

1) A major result of the manuscript is that Tsimane' and Western listeners had similar patterns of results on the fusion task. It is good to demonstrate and quantify this fact, of course, but wasn't the outcome to be expected? If Tsimane' were not able to group sounds based on harmonicity, would they not fail to hear e.g. vowels as single sounds, or extract an harmonic target from a noisy background. Is there an underlying hypothesis that could have predicted an alternative outcome on this fusion task?

2) A second major result is that, unlike other listeners, Tsimane' did not prefer fused intervals. This is very interesting, and interpreted as showing that there is common machinery to process fusion, but that fusion's aesthetic value is acquired. But, is it possible to rule out another interpretation, namely that Stumpf's theory of consonance is simply wrong? Here, the authors have found a dissociation between the perception of fusion and the perception of consonance. Note also that the Parncutt review cites an ethnomusicology paper claiming the opposite dissociation, very dissonant sounds which are fused (Kaminski, *Muzyka*, 2009). How can we rule out that in fact the two notions are independent, and only fortuitously associated through exposure to Western music?

3) The last point, perhaps a finer argument, is that consonance itself is not reducible to pleasantness, as already acknowledged by the authors. Even if I can easily buy the operational definition proposed in the manuscript, it could perhaps still be acknowledged somewhere that maybe there is no equivalent concept in Tsimane' music.

Specific comments

L270, L333: There is a preference of Tsimane' listeners for high root pitch. This seems intriguing, and perhaps suggests that for Tsimane' the preference judgments are not entirely driven by what we would call consonance. What could underlie this preference? A recognition of playing effort? Higher prevalence of high pitches in Tsimane' music?

L305: "the basic perceptual machinery for music is similar across cultures." It is really a matter of wording, but a more neutral statement would be that the machinery for sound is similar across cultures. I do not think that the perception of fusion is specific to music.

L500: Tsimane' use pentatonic scales, so presumably they include a few of the intervals tested?

L608: No minor or major thirds were included in the experiments because they did not have more fusion than dissonant interval in Western listeners. Isn't this a further problem for the theory of consonance as fusion, given that both intervals are considered consonant in Western music?

L636: There is a big difference in the training phase for Tsimane' compared to Western listeners, even though this phase involved control tones. Maybe comment?

L828. Typo in the Oomph in Stumpf (a shame with such a title!).

Axes labels for Supplementary Figure 4 do not match.

Response to Reviews – McPherson et al.

Overview

The reviewers identified a design flaw in our original experiments: the inclusion of conditions with feedback that could in principle have “trained” participants to respond in a particular way on our fusion tasks. We originally ran the experiments with feedback for fear that the task would otherwise be ambiguous, but in retrospect this was a poor choice. We thus returned to Bolivia and ran a new set of experiments with a new set of Tsimane’ participants, using a different design in which no feedback was provided (we also collected a new companion data set from Boston non-musicians). We were pleased to see that the results were very similar in the absence of feedback. We have thus replaced the original experiments with the new ones.

In addition, we extended the general phenomena by 1) conducting analogous experiments with sung tones, 2) including two new control experiments, 3) conducting an online experiment with large N to compare fusion and consonance in Westerners with high precision, and 4) conducting an analysis of individual differences. The main results thus now represent a significant advance over our 2016 paper.

The reviews were perceptive and helpful. We have revised the paper taking them into account, and believe it is much improved. Below we provide a point-by-point response to each comment from the reviews, describing the changes that have been made to address each comment.

Reviewer #1 (Remarks to the Author):

This manuscript describes the results of a study that follows up on the earlier study of McDermott et al. (2016; *Nature*, 535, 547-550). The earlier study demonstrated that Western concepts of musical consonance and dissonance were not universal by showing that members of the Tsimane in a remote region of Bolivia did not express a preference for consonant over dissonant musical intervals. This interesting finding was bolstered by showing that the Tsimane were able to discriminate harmonic from inharmonic stimuli – they just didn’t “prefer” the harmonic over the inharmonic, in contrast to the judgments of Western listeners.

There is much positive about this new contribution. It is well written and provides a very clean replication of the original findings concerning a lack of preference for consonant over dissonant musical intervals. This is reassuring and helps strengthen the original claims of the authors. The main claim to novelty of the current submission is the finding that the Tsimane can distinguish to some extent dissonant from consonant intervals. This is demonstrated by showing that both Tsimane and Westerners (Boston non-musicians) are more likely to rate consonant than dissonant intervals as being one sound, rather than two. Although consistent with earlier findings, this new study does not provide much that is novel or scientifically significant.

The main finding, that Tsimane can distinguish harmonic from inharmonic sounds, was already reported in the original 2016 study (their Fig. 4). The difference here is that musical intervals, rather than artificially mistuned tones, are used, and that the judgement is “one tone” vs. “two tones”, notionally targeting the concept of fusion. However, all the listeners were first trained with feedback on what “one tone” and “two tones” are supposed to sound like, and the training stimuli were not all that different from the ones used in the main experiment. As shown by the authors, the “consonant” intervals used (octave, fifth, fourth) are all good approximations to a single harmonic series, with (in the case of fifth and fourth) just a few missing harmonics. Because the tones were synthesized to have identical onsets and temporal envelopes, there is not much in the way of acoustic cues to indicate that there are actually two sounds present. In contrast, the “dissonant” intervals clearly do not follow a single harmonic series and are in that way much more like the inharmonic sound used in training (which is, in fact, a musical minor 9th – one of the stimuli used in the main experiment). Thus, it is not clear how the work extends beyond the original conclusion that the Tsimane (and by extension humans in general) can distinguish harmonic from inharmonic sounds.

Thank you for the thoughtful critique. In hindsight we agree, and felt this issue was sufficiently important that it merited rethinking the design. Last summer we returned to Bolivia and ran new experiments omitting the training conditions. These training conditions, and the associated feedback, turned out to be completely unnecessary – we obtained similar results simply by presenting pairs of notes and asking participants if they heard one or two sounds. We believe this result goes substantially beyond what we showed in our earlier paper, to indicate the likelihood that the perception of fusion is qualitatively similar across cultures, and that it cleanly dissociates from musical interval preferences.

Indeed, as it has previously been shown that this ability extends to other species (e.g., Izumi, 2000, JASA; Hulse et al., 1995, JEP:General), it would be particularly surprising if the Tsimane were unable to perform this distinction with training. The fact that over 60% of the remote Tsimane needed to repeat the training/screening session because they initially scored under 60% correct suggests that the distinction based on perceived fusion was not particularly natural, even if they could eventually perform it.

We agree that cross-species results are relevant, and have included references to studies in non-human animals that showed an ability to discriminate harmonic and inharmonic sounds. We also now discuss what our experiments show above and beyond those earlier studies.

This problem also relates to the title of the manuscript: The ability to distinguish harmonic from inharmonic stimuli does not necessarily imply anything about consonance and dissonance, which by definition are subjective labels that have already been shown not to be universal. It is therefore far from clear what the authors mean to imply by claiming “universal representations of consonance and dissonance.”

We agree, and have changed the title to avoid conflating harmonic and inharmonic with consonance and dissonance.

Other than this crucial major concern, there are some minor issues that should be addressed, as follows:

Line 47. The references to papers on detection of mistuned harmonics are unfortunate here, as there are counter- examples (e.g., Roberts and Brunstrom, 1998, JASA) showing that components mistuned from a regularly spaced but inharmonic complex tone can also be heard out.

We are aware of these findings and appreciate them, but feel they speak to the mechanism underlying harmonicity-based sound segregation, as opposed to undermining the idea that harmonicity is important for segregation. The Roberts results are important, but we are inclined to leave discussion of this to other papers more directly concerned with the mechanisms of harmonicity-based segregation, as the issue seems peripheral to the main argument in the paper.

Line 64. Pick f_0 or F_0 and then stick to it.

Done.

Lines 65-67. I don't know what remains "unclear" here. The results of McDermott et al. (2016, Nature) are reasonably clear on this question: inharmonicity can be detected by non-Western listeners, but it isn't necessarily judged as more unpleasant than harmonicity.

This sentence has been removed as part of the revisions to the introduction, which we hope now more clearly specifies what was left open by prior work.

The relevant parts of the introduction now read:

"Irrespective of its relationship to consonance, harmonicity-based sound segregation might provide an important constraint on musical systems, particularly if its perceptual effects on musical note combinations were present cross-culturally. A priori, it seemed plausible that this might be the case, but not inevitable. The measurements of fusion that exist have been limited to Western listeners, who have extensive exposure to harmony featuring simple-integer-ratio intervals, and for whom fusion could thus reflect learned "schemas"^{5,41,42}, potentially incorporating the idiosyncrasies of modern tuning systems. Moreover, sound segregation abilities are often thought to change with (Western) musical training⁴³⁻⁴⁶, which might suggest that the phenomenon of fusion could vary across cultures differing in their musical experience. We thus sought to verify the phenomenon of fusion in Westerners, test its robustness to tuning systems, test its relation to consonance in Westerners, and explore the extent to which it is present cross-culturally."
(lines 87-100)

"...In that earlier study we found that the Tsimane' could detect mistuned harmonics⁴⁰, suggesting some sensitivity to harmonicity, but their overall sensitivity was worse than the comparison group of Westerners, and the experiment did not involve actual musical intervals (pairs of notes). It was thus unclear whether their perceptual representations of musical intervals (as measured by fusion) would qualitatively differ along with their aesthetic evaluations, or whether their representations of note combinations would resemble those of Westerners despite these differences in aesthetic evaluations."
(lines 111-118)

Line 126. None of the cited references mistuned all the even from the odd

harmonics. There are papers that have done this, but the cited references only mistuned a single component.

We have omitted the condition with mistuned even harmonics from the experiments in the revised paper.

Lines 130-134. This is problematic experimentally. The only sounds that could result in feedback telling participants that they were incorrect were the inharmonic sounds, if the participants responded “one sound”. Assuming that listeners can distinguish harmonic from inharmonic (as demonstrated by McDermott et al., 2016), it seems reasonable that listeners would quickly learn to respond “two sounds” to the inharmonic stimuli based on the training feedback. Although the claim that harmonic sounds are more “fused” is very reasonable, the paradigm is somewhat flawed and in danger of being circular.

We agree, and ran new experiments removing all feedback, and these conditions in particular. It turns out not to have been important.

Line 339. This statement is in danger of being perceived as severely disingenuous. The 2014 paper cited here throws serious doubt on earlier claims that infants look longer (or “prefer”) consonant over dissonant intervals, and yet it is cited here as if it is one of the earlier studies that reports just such a finding.

We now provide a more nuanced description of this literature, citing the 2014 paper separately and making it clear that it suggests a different conclusion:

“Our findings could help explain previous findings in Western infants, who have in some cases been reported to produce longer gaze durations for consonant compared to dissonant intervals^{59,60} (though not always⁶¹, and although some such results were plausibly driven by roughness⁵⁹, which we find to be aversive even to the Tsimane’, and which does not predict consonance preferences in Westerners²⁷).”
(lines 394-399)

Lines 412-413. It is not necessary to report mean age or years of schooling to 4 significant figures.

Corrected.

Line 466. Given the discrete nature of many of the variables, wouldn't a Spearman's rank-order correlation be more appropriate?

Given the number of additional experiments in this revision, we have removed the acculturation analysis to focus on the fusion results, so this is no longer an issue.

Supplementary Figure 3a. Typo in heading (judgmeents).

This figure has been removed from the revised manuscript.

Reviewer #2 (Remarks to the Author):

The work examines the ability of listeners naïve to Western music to distinguish two-note chords based on major or minor intervals from single sounds. The work represent an almost unique examination of listeners without Western music exposure who will likely become rarer based on the shifting demographic of this group. The more Western listeners were better at detecting the presence of two-note chords overall. But the key finding is that the naïve listeners demonstrated the same effect of whether the note combinations were consonant or dissonant on detecting note pairs despite the fact that they did not recognise these intervals.

The data support a difference in the fundamental encoding of consonant and dissonant chords which is plausibly explained by similar sensitivity to the harmonicity of complex sounds in the different groups despite the different encultured experience of major and minor keys. I am not surprised by this, considered from an auditory neuroscience perspective: the two- versus one-note distinction can be explained based on harmonicity which is a strong behavioural grouping cue and for which hard-wired neurophysiological bases have been demonstrated in animal models. But considered from a musical perspective it is important new information about the basis for learned consonance versus dissonance which cannot be attributed to differences in fundamental sensory representation and grouping.

Thank you. We believe this finding is strengthened by the new experiments.

Reviewer #3 (Remarks to the Author):

The manuscript reports a set of behavioral experiments that compared musical interval perception in Western listeners and in Tsimane' listeners. The aim is a cross-cultural comparison of the perception of 'fusion' and 'consonance' for listeners with a varying degree of familiarity with Western music. Two main tasks were run, in both groups: a fusion task, for which listeners had to report whether they heard a single tone or two tones for various intervals, and a consonance task, for which listeners rated the pleasantness of these same intervals. All groups of listeners showed a greater tendency to report a single sound, so more fusion, for consonant intervals. However, there was no preference for consonant intervals in Tsimane' listeners, unlike for Western listeners and even to some extent for Westernized Tsimane' listeners.

This study is a new installment in a rich series of experiments comparing Tsimane' and Western listeners for various perceptual task, using musical or music-related material. As with previous papers, the question asked is relevant and interesting, the experimental procedures are impressive and solid, and the dataset is highly valuable and unique. An added twist is that there are now correlations found with the degree of Westernization of Tsimane' listeners, underscoring the timeliness of such comparative studies. Finally, the manuscript is very well written and the results clear-cut. As such, I have very little remarks for improvements directly related to my core field of expertise.

Thank you.

The only set of questions I would like to raise is about the link between fusion and consonance. Fusion is a concept from experimental psychology, whereas of course consonance has a long musical history. As underlined in the manuscript, a link between the two has been suggested 150 years ago by Stumpf. But, since then, such a link has been debated (for instance, see the recent review by Parncutt and Hair, *Journal of Interdisciplinary Music Studies*, 2011). Reading the manuscript, I am not fully sure whether the authors have shown a common basis for consonance judgments, or rather brought strong contradictory evidence against Stumpf's proposal.

We largely agree, and have revised the manuscript with this perspective in mind.

General comments

1) A major result of the manuscript is that Tsimane' and Western listeners had similar patterns of results on the fusion task. It is good to demonstrate and quantify this fact, of course, but wasn't the outcome to be expected? If Tsimane' were not able to group sounds based on harmonicity, would they not fail to hear e.g. vowels as single sounds, or extract an harmonic target from a noisy background. Is there an underlying hypothesis that could have predicted an alternative outcome on this fusion task?

It seems reasonable to expect that segregation of speech sounds would be similar across cultures, but we think it was highly nonobvious a priori whether patterns of segregation of musical notes would be similar as a result, particularly given 1) the rather striking differences we have seen between US and Tsimane' participants in many other aspects of music perception and 2) the substantially worse sensitivity to inharmonicity we found previously in the Tsimane'. We note also that there are many reports of musical experience affecting the ability to segregate sounds (refs 43-46 in the text), which might be taken to suggest that sound segregation could vary across cultures. We have revised the introduction to better motivate the experiments and explain why it seemed possible a priori to obtain a different result:

“Irrespective of its relationship to consonance, harmonicity-based sound segregation might provide an important constraint on musical systems, particularly if its perceptual effects on musical note combinations were present cross-culturally. A priori, it seemed plausible that this might be the case, but not inevitable. The measurements of fusion that exist have been limited to Western listeners, who have extensive exposure to harmony featuring simple-integer-ratio intervals, and for whom fusion could thus reflect learned “schemas”^{5,41,42}, potentially incorporating the idiosyncrasies of modern tuning systems. Moreover, sound segregation abilities are often thought to change with (Western) musical training⁴³⁻⁴⁶, which might suggest that the phenomenon of fusion could vary across cultures differing in their musical experience. We thus sought to verify the phenomenon of fusion in Westerners, test its robustness to tuning systems, test its relation to consonance in Westerners, and explore the extent to which it is present cross-culturally.”
(lines 87-100)

“...In that earlier study we found that the Tsimane' could detect

mistuned harmonics⁴⁰, suggesting some sensitivity to harmonicity, but their overall sensitivity was worse than the comparison group of Westerners, and the experiment did not involve actual musical intervals (pairs of notes). It was thus unclear whether their perceptual representations of musical intervals (as measured by fusion) would qualitatively differ along with their aesthetic evaluations, or whether their representations of note combinations would resemble those of Westerners despite these differences in aesthetic evaluations.”
(lines 111-118)

2) A second major result is that, unlike other listeners, Tsimane’ did not prefer fused intervals. This is very interesting, and interpreted as showing that there is common machinery to process fusion, but that fusion’s aesthetic value is acquired. But, is it possible to rule out another interpretation, namely that Stumpf’s theory of consonance is simply wrong? Here, the authors have found a dissociation between the perception of fusion and the perception of consonance. Note also that the Parncutt review cites an ethnomusicology paper claiming the opposite dissociation, very dissonant sounds which are fused (Kaminski, Muzyka, 2009). How can we rule out that in fact the two notions are independent, and only fortuitously associated through exposure to Western music?

We think this interpretation is largely correct, given some important qualifications. To further test this idea, we made two additions to the revised set of experiments.

First, in our new in-person fusion/preference experiment we included the major third. We find the major third to be less fused than other consonant intervals for both Tsimane’ and Boston participants, even though the Boston participants rate it highly in terms of pleasantness (Figure 3 in the revised manuscript).

Second, to better substantiate the dissociation between fusion and consonance preferences suggested by the in-person experiment, we conducted an online experiment with a large number of participants and a larger interval set. The large sample size enabled highly reliable estimates of the fusion and pleasantness of individual intervals, and revealed systematic differences between the two: the most fused intervals are not as pleasant as would be expected given the extent of their fusion, and the other consonant intervals are more pleasant than would be expected given the relatively weak fusion they produce (Figure 4a-c in the revised manuscript).

In addition, the large sample size in the online experiment enabled us to examine individual differences. We summarized the consonance-dissonance fusion effect as the difference in fusion between consonant and dissonant intervals, and the preference for consonance as the difference in pleasantness between consonant and dissonant intervals. These two measures exhibited reliable differences across individuals, but themselves were not correlated (Figure 4d in the revised manuscript). This suggests that the fusion effect does not directly cause an individual's preference for consonance, as might be expected if such preferences resulted from associating fused intervals with positive valence.

These results underscore the idea that the preferences of Westerners are not straightforwardly determined by the perceptual constraints of harmonicity-based fusion. However, the fact remains that the phenomenon of fusion is present cross-culturally, and picks out certain musical intervals as special, and that these intervals overlap with those considered consonant by Westerners (and with those prominent in other musical systems). Fusion thus seems to have plausibly played a role in the historical evolution of consonance in Western music, and in the broader development of musical systems around the world. To our knowledge our results provide the first clear evidence for a universal perceptual phenomenon that could shape the way intervals are used in music, even though we agree that they also show that Western consonance (as measured with pleasantness) is clearly affected by other factors as well (also evident in fluctuations in definitions of consonance throughout the history of Western music). We have thoroughly revised the paper to make this point explicit.

3) The last point, perhaps a finer argument, is that consonance itself is not reducible to pleasantness, as already acknowledged by the authors. Even if I can easily buy the operational definition proposed in the manuscript, it could perhaps still be acknowledged somewhere that maybe there is no equivalent concept in Tsimane' music.

We agree, and have included a note to this effect. We maintain that pleasantness remains the best operational definition of consonance, particularly in the context of experiments with musically untrained participants, for whom the word "consonant" will not have a clear meaning, and in the context of cross-cultural experiments, where "consonant" is not straightforward to translate (McDermott et al. 2016).

The revised text reads:

“We operationalized consonance as pleasantness because lay listeners might not be familiar with the term “consonance”, and because it was impossible to translate into Tsimane’. We are cognizant that consonance is a multifaceted phenomenon in the context of music theory^{18,39}. Here we assessed pleasantness by querying “liking” for ease of translation. We have found previously that the liking task with four response choices (described above) yields similar results to a pleasantness rating scale task^{27,40}.”
(lines 620-626)

Specific comments

L270, L333: There is a preference of Tsimane’ listeners for high root pitch. This seems intriguing, and perhaps suggests that for Tsimane’ the preference judgments are not entirely driven by what we would call consonance. What could underlie this preference? A recognition of playing effort? Higher prevalence of high pitches in Tsimane’ music?

Given the new experiments and altered focus of the manuscript, we no longer comment on this finding in the revised manuscript. However, the preference for higher pitch has been present in every experiment we have ever conducted with Tsimane’ participants. We have asked them about why they prefer higher pitches and they are aware of their preference, and have described higher pitched sounds as “clearer”. It is hard to know what causes this effect (or what causes the preference for lower pitches that is widespread among US participants). We note that the most common Tsimane’ instrument is the flute, which has a relatively high pitch range, but can only speculate as to whether this relates to the preference that they exhibit.

L305: “the basic perceptual machinery for music is similar across cultures.” It is really a matter of wording, but a more neutral statement would be that the machinery for sound is similar across cultures. I do not think that the perception of fusion is specific to music.

We have replaced this with a more nuanced statement:

“...the basic perceptual machinery for sound segregation is similar across cultures, producing shared representations of musical

**structure...”
(lines 357-359)**

L500: Tsimane’ use pentatonic scales, so presumably they include a few of the intervals tested?

Yes, although their traditional music is mostly monophonic. So they presumably are familiar with melodic intervals, less so with harmonic intervals. We now mention that this property of their music means they likely had some familiarity with some of the intervals we used, at least in melodic contexts:

**“Tsimane’ participants thus may have had prior familiarity with melodic versions of some of the intervals used in the present experiments, but likely much less so with the harmonic versions that were actually used.”
(lines 574-577)**

L608: No minor or major thirds were included in the experiments because they did not have more fusion than dissonant interval in Western listeners. Isn’t this a further problem for the theory of consonance as fusion, given that both intervals are considered consonant in Western music?

Yes. In the new experiments we included the major third, to demonstrate this effect and explore its origin. Neither Boston nor Tsimane’ listeners showed a substantial fusion increase for the third, despite it being rated as among the most pleasant intervals by Western listeners (evident in our Boston data in this paper). We now interpret this result, along with those of the new online experiment, to indicate that consonance is not simply “fusion plus valence”.

L636: There is a big difference in the training phase for Tsimane’ compared to Western listeners, even though this phase involved control tones. Maybe comment?

This is no longer relevant now that we have run a new experiment that does not involve a training phase, or feedback of any sort. Instead we ran an additional experiment to verify experiment comprehension, in which participants heard segments of speech (either one person speaking, or two concurrent talkers) and judged whether they heard one or two sounds. The Tsimane’ all performed well at this task.

L828. Typo in the Oomph in Stumpf (a shame with such a title!).

Corrected.

Axes labels for Supplementary Figure 4 do not match.

This has been corrected.

Reviewers' comments:

Reviewer #1 (Remarks to the Author):

The authors have addressed the major concerns of the reviewers in terms of potential flaws in the experimental designs. This remains a very well-written and informative paper. The main weakness is still that the results are basically predictable, given what we already know about the universality of harmonic representations. Indeed, it is hard to imagine how sensitivity to harmonicity would not be shared by all humans, given that we know it is present in other species. Nevertheless, the experiments have been well designed and executed and will be of interest, especially given the trouble the authors went to in collecting the data. Although the paper does not represent a major conceptual advance and does not reshape our thinking in this field, it will nevertheless be a useful addition to the literature.

Reviewer #2 (Remarks to the Author):

The authors have dealt well with the issues raised by all the reviewers and the manuscript is suitable for publication

Reviewer #3 (Remarks to the Author):

The revision of the manuscript is quite an extensive one. The authors have collected a whole new dataset, mainly controlling for a potential confounding factor in the first version of the experiments, but also adding a set of useful controls and baselines. The results and conclusions still hold. This is all the more reassuring as we now have essentially an extension plus a replication of the study. Given the difficulty of obtaining such data in Tsimane' listeners, this is really to be commended. The authors have also added a completely new online experiment, addressing an important issue on the link between fusion and consonance in Western listeners. This experiment has not been subjected to peer review yet, so I tried in my remarks below to focus on its results and interpretations.

My main issue with the first version of the manuscript was with the framing of the results in terms of fusion vs consonance. On this point, I should stress that I am fully convinced by the rewrite of the Introduction and Discussion, as well as with the new title.

Specific comments

L181-210, L218, etc. The presentation of the statistical tests was a bit disconcerting at first, as parametric and non-parametric tests seem to be used with no apparent logic. There are Wilcoxon tests, t-tests, Wilcoxon tests complemented by what seem to be standard ANOVAs... The Methods on L852 provide a justification for these choices: an apparent deviation from normality for some parts of the dataset and not others. If the stats are to be kept as is, I would suggest to move the explanation to the

main text. But, as it seems that normality was assessed “by eye” (L853), why not use only non-parametric tests? Or only parametric ones, as t-tests are supposedly quite robust in the face of violations of normality? In any case, at least the presentation of the stats should be improved.

L348. The results of the fusion experiments are compared to what is “roughly” predicted by a model of similarity with harmonic series. Why not use an actual model, one from the literature or a new one fitted to the new and extensive dataset available here? This is an optional suggestion that the authors should feel free to disregard altogether.

Figure 4 is new, with the big claim that Western listeners show a dissociation between consonance and fusion. I have a few questions on the analyses supporting the claim.

- on panel c, there is an indication of a correlation coefficient, r . Is it r on the mean data, a mean of r on individual data? Wouldn't it be more meaningful to estimate a rank-order correlation, as the two scales of Pleasantness and Fusion ratings are a priori unrelated?

- for the same panel, the correlation actually looks pretty tight when the octave is disregarded together with the unison. Is this actually the case?

- if so, the individual test-retest analyses of panel d would really be key to argue that consonance and fusion are dissociated. After a few re-reads, I think I get the idea: even though consonance and fusion are highly correlated on average, they are not consistently related in individual participants. This is not because of noisiness in the data as test-retest reliability is high for individual measures. Is this correct? In any case, it may be useful to develop the rationale and outcome of this analysis in the main text, especially as the conclusion seems to run against the visual impression given by panel c.

- is it correct that the octave and unison were excluded from the analyses of panel d?

L596. Methods. This is really to be picky, but how do you calibrate a sound delivery system by connecting an artificial ear to an audiometer?

L 868. Repeated description of the Mauchly test.

Reviewer #1 (Remarks to the Author):

The authors have addressed the major concerns of the reviewers in terms of potential flaws in the experimental designs. This remains a very well-written and informative paper. The main weakness is still that the results are basically predictable, given what we already know about the universality of harmonic representations. Indeed, it is hard to imagine how sensitivity to harmonicity would not be shared by all humans, given that we know it is present in other species. Nevertheless, the experiments have been well designed and executed and will be of interest, especially given the trouble the authors went to in collecting the data. Although the paper does not represent a major conceptual advance and does not reshape our thinking in this field, it will nevertheless be a useful addition to the literature.

Thank you.

Reviewer #2 (Remarks to the Author):

The authors have dealt well with the issues raised by all the reviewers and the manuscript is suitable for publication

Thank you.

Reviewer #3 (Remarks to the Author):

The revision of the manuscript is quite an extensive one. The authors have collected a whole new dataset, mainly controlling for a potential confounding factor in the first version of the experiments, but also adding a set of useful controls and baselines. The results and conclusions still hold. This is all the more reassuring as we now have essentially an extension plus a replication of the study. Given the difficulty of obtaining such data in Tsimane' listeners, this is really to be commended.

Thank you.

The authors have also added a completely new online experiment, addressing an important issue on the link between fusion and consonance in Western listeners. This experiment has not been subjected to peer review yet, so I tried in my remarks below to focus on its results and interpretations.

My main issue with the first version of the manuscript was with the framing of the results in terms of fusion vs consonance. On this point, I should stress that I am fully convinced by the rewrite of the Introduction and Discussion, as well as with the new title.

Thank you.

Specific comments

L181-210, L218, etc. The presentation of the statistical tests was a bit disconcerting at first, as parametric and non-parametric tests seem to be used with no apparent logic. There are Wilcoxon tests, t-tests, Wilcoxon tests complemented by what seem to be standard ANOVAs... The Methods on L852 provide a justification for these choices: an apparent deviation from normality for some parts of the dataset and not others. If the stats are to be kept as is, I would suggest to move the explanation to the main text. But, as it seems that normality was assessed “by eye” (L853), why not use only non-parametric tests? Or only parametric ones, as t-tests are supposedly quite robust in the face of violations of normality? In any case, at least the presentation of the stats should be improved.

Some of the fusion experiment results distributions deviated markedly from being normal, and it is usually considered best practice to avoid tests that assume normality in such conditions. We thus use non-parametric tests in all cases involving fusion results.

We used parametric tests for the pleasantness rating results because they were approximately normal and because parametric tests should maximize power. This seemed worthwhile given that these experiments produced null effects in some cases for the Tsimane’ (unlike the fusion experiments). However, we confirmed that the results of all t tests (i.e., whether they were above or below the standard .05 threshold) remained the same if Wilcoxon tests were used instead.

We agree that this merited clarification, and have added a sentence to the Results section explaining why non-parametric tests were used for the fusion results:

“Non-parametric tests were used to evaluate all fusion experiments as the data were non-normal.” (p. 9, line 22)

In addition, we have clarified in the Results section that the significance of the F statistic was determined using a non-parametric test rather than a standard ANOVA:

“Pooling across experiments with synthetic and sung tones, there was no interaction between participant group and interval type ($F(1,57)=3.42$, $p=.07$, $\eta_p^2=.057$, significance evaluated via bootstrap due to non-normality).” (p. 12, lines 1-3)

Finally, in the Statistics section of the Methods we have more clearly delineated the approaches used for the fusion and pleasantness rating experiments:

“Data distributions were evaluated for normality by visual inspection. The

data from the fusion experiments were often non-normal due to the prevalence of response proportions near 0 and 1, and so we relied on non-parametric tests. Significance of pairwise differences between fusion conditions was evaluated using a Wilcoxon signed-rank test. For multi-condition tests of fusion judgments (analogous to ANOVAs), we computed F statistics and evaluated their significance with approximate permutation tests, randomizing the assignment of the data points across the conditions being tested 10,000 times.

For the preference experiments, paired t-tests were used to test for differences between conditions within groups, and mixed-design ANOVAs were used to test for main effects of stimulus condition and for interactions between the effect of stimulus condition and participant group.”
(p. 44, line 14 – p. 45, line 5)

L348. The results of the fusion experiments are compared to what is “roughly” predicted by a model of similarity with harmonic series. Why not use an actual model, one from the literature or a new one fitted to the new and extensive dataset available here? This is an optional suggestion that the authors should feel free to disregard altogether.

We accept the invitation to put modeling off to a future paper. As we allude to in the introduction, the field lacks models of the perception of harmonicity that are validated and consistent with what we know about human pitch perception and/or harmonicity-based sound segregation. Part of the difficulty is that the perception of harmonic sounds depends strongly on the rank of the harmonics in the sound (i.e., whether the harmonic is “resolved” by the cochlea). Thus, simple measures based on the autocorrelation or harmonic templates applied to the power spectrum are inadequate. This is an important direction for future research, but beyond the scope of this paper. We have added a sentence to the introduction to help explain why current models are not that helpful in this context:

“...the extent to which consonance can be fully predicted by similarity to the harmonic series remains unclear, in part because we lack widely accepted models for how harmonicity is represented in the auditory system^{23,25,30}. In particular, the perception of harmonicity is strongly influenced by the position of harmonics within the harmonic series³⁰, and is not well captured by naive measures based on harmonic templates or autocorrelation.” (p. 4, lines 5-11)

Figure 4 is new, with the big claim that Western listeners show a dissociation between consonance and fusion. I have a few questions on the analyses supporting the claim.

- on panel c, there is an indication of a correlation coefficient, r . Is it r on the mean data, a mean of r on individual data?

Yes, that correlation is on the mean data. We have clarified this in the text:

“When directly compared, mean consonance and mean fusion for Westerners were correlated across intervals” (p. 16, lines 16-17)

And in the figure caption:

“ r value is the correlation between the mean fusion and mean pleasantness across intervals.”

And in the title of the results graph, which now specifies that it is plotting the mean results for each interval:

c. Mean Fusion vs. Mean Pleasantness For Each Interval

Wouldn't it be more meaningful to estimate a rank-order correlation, as the two scales of Pleasantness and Fusion ratings are a priori unrelated?

We have replaced Pearson correlations with Spearman here and elsewhere. Results/conclusions are qualitatively unchanged.

- for the same panel, the correlation actually looks pretty tight when the octave is disregarded together with the unison. Is this actually the case?

The correlation increases without the octave (from $r_s = .85$ to $r_s = .93$), as you would expect from the scatter plot. However, we don't see a good justification for excluding the octave. Instead, we have included all data from the study as a supplementary file, so that readers can explore questions like this.

We also now note that although the correlation appears high relative to what one often encounters in biology and psychology, it is much lower than it could be given the reliability of the measurements. Because of the large sample, the patterns of mean fusion and mean pleasantness are each highly reliable ($r_s = .99$ and $.98$), such that the correlation between them could be as high as $.99$. The discrepancies reflected in the actual correlation are thus real and meaningful. We have clarified this in the revised text:

“...the correlation between mean consonance and fusion was lower than it could have been given the high reliability of the mean results ($r_s=.99$ and $.98$ for fusion and consonance, respectively).” (p. 17, lines 3-5)

- if so, the individual test-retest analyses of panel d would really be key to argue that consonance and fusion are dissociated. After a few re-reads, I think I get the idea: even though consonance and fusion are highly correlated on average, they are not consistently related in individual participants. This is not because of noisiness in the data as test- retest reliability is high for individual measures. Is this correct? In any case, it may be useful to develop the rationale and outcome of this analysis in the main text, especially as the conclusion seems to run against the visual impression given by panel c.

Your understanding of what we did, and the underlying logic, is correct. If fusion underlies consonance (here operationalized as pleasantness), and if there are reliable individual differences in fusion and pleasantness, then individuals with larger fusion differences between consonant and dissonant intervals should have larger pleasantness differences. So we first measured the reliability of the individual differences by splitting the data for each participant in two halves, to make sure there were reliable individual differences. We then compared fusion and consonance in individuals. The results suggest that fusion does not cause consonance. We have clarified what we did, and the logic behind it, in the revised text:

“The large sample size also enabled analysis of individual differences in fusion and consonance preferences. As shown in Fig. 4d and 4e, consonant and dissonant intervals differ on average in fusion and pleasantness, but the extent of these effects varies from person to person. If fusion directly causes consonance preferences (i.e., if listeners prefer sounds that are more fused), then individuals with large fusion effects should also have large preference effects provided the individual differences are reliable.” (p. 17, lines 8-14)

and

“Both fusion and consonance preferences exhibited reliable individual

differences – participants with large effects on one half of the trials (split 1) also had large effects on the other half of the trials (split 2) (Fig. 4d and 4e, right panels; $r_s=.49$, $p<.001$ and $r_s=.85$, $p<.001$ for fusion and consonance, respectively, Spearman-Brown corrected, $p<.001$ in both cases). These reliable individual differences should be correlated if the two effects are related. However, the two effects were not correlated across participants (Fig. 4f, right panel; $r_s=.01$, $p=.93$.” (p. 17, line 21 – p. 18, line 5)

We also revised Figure 4 to make the analysis more explicit:

- is it correct that the octave and unison were excluded from the analyses of panel d?

No. The analyses use all the intervals other than the unison, pooling all the consonant intervals (those colored blue in 4a and 4b) and all the dissonant intervals (those colored brown) and taking their difference. This has been clarified in the main text as well as in the figure caption and figure. As noted in the text and shown in Supplementary Figure 2, we obtained similar results using smaller subsets of intervals, so the results do not depend sensitively on those choices.

New description in the main text:

“We used all the intervals in the experiment apart from the unison

(consonant and dissonant interval sets were those colored blue and brown, respectively, in Fig. 4a and 4b).” (p. 17, lines 17-19)

Excerpt from revised figure caption:

“Consonant and dissonant intervals were all those colored blue and brown, respectively, in a-c”

We also list the intervals used in the revised Figure 4.

L596. Methods. This is really to be picky, but how do you calibrate a sound delivery system by connecting an artificial ear to an audiometer?

The artificial ear simulates the acoustics of the human ear canal, and the microphone is supposed to record the sound that would be produced at the eardrum were the headphones on the head of a participant. It is a standard method for calibration, allowing one to measure the headphone output as it would be coupled to the ear of the listener. We have clarified this in the Methods:

“The audio presentation system was calibrated ahead of time with a GRAS 43AG Ear & Cheek Simulator connected to a Svantek’s SVAN 977 audiometer. This setup is intended to replicate the acoustic effects of the ear, measuring the sound level expected to be produced at the eardrum of a human listener, enabling tone presentation at the desired sound pressure level.” (p. 31, line 21 – p. 32, line 3)

L 868. Repeated description of the Mauchly test.

Corrected.

***REVIEWERS' COMMENTS:

Reviewer #3 (Remarks to the Author):

The minor revision to the text and the changes to Figure 4 are sufficient to address my previous comments. I would like to congratulate the authors for an impressive piece of work.

Reviewer's Comments:

Reviewer #3 (Remarks to the Author):

The minor revision to the text and the changes to Figure 4 are sufficient to address my previous comments. I would like to congratulate the authors for an impressive piece of work.

Thank you.